# Critical behavior of a phase transition in the dynamics of interacting populations

Thibaut Arnoulx de Pirey[1] and Guy Bunin[2]

**1** Paris, France
**2** Department of Physics, Technion-Israel Institute of Technology, Haifa 32000, Israel
* thibautdepirey@gmail.com

April 17, 2024

## Abstract

**Many-variable differential equations with random coefficients provide powerful models for the dynamics of many interacting species in ecology. These models are known to exhibit a dynamical phase transition from a phase where population sizes reach a fixed point, to a phase where they fluctuate indefinitely. Here we provide a theory for the critical behavior close to the phase transition. We show that timescales diverge at the transition and that temporal fluctuations grow continuously upon crossing it. We further show the existence of three different universality classes, with different sets of critical exponents, depending on the migration rate which couples the system to its surroundings.**

# 1 Introduction

Many high-dimensional dynamical systems, of interest in neuroscience [1], ecology [2–4], game theory [5] and economics [6], exhibit two dynamical phases, one where the dynamics reach a fixed point, and another where the variables fluctuate indefinitely. A sharp phase transition between these phases is observed when varying system parameters. In many cases, this transition has been linked to a loss of fixed point stability and the emergence of unstable fixed points whose number is exponentially large in the dimension of the system [7–9].

The first instance where such a transition was predicted are models of high-dimensional random neural networks [1], which are by now well-understood [10]. Here, as for the p-spin dynamics [11], the dynamics of a single degree of freedom become Gaussian in the high-dimensional limit and closed equations for two-time correlation functions can then be derived [1,12]. In the fluctuating phase, the dynamics reach a chaotic time-translation invariant state [1]. The transition is of second order, with a continuous growth of the amplitude of time fluctuations when going away from the critical point. The critical behavior has been investigated and the critical exponents governing the growth of fluctuations and timescales have been obtained for various types of nonlinearities [13].

Species-rich ecosystems can also be modeled by high-dimensional nonlinear dynamics for the different species population sizes. Common ecological models, such as Lotka-Volterra or resource-competition models [14], with random interactions between species, are also known to exhibit a phase transition from fixed point to persistent fluctuations [2, 3]. The Lotka-Volterra dynamics are a standard model for a well-mixed system (no spatial extension). The dynamics of the population sizes $N_i$ of the species $i = 1 \ldots S$, with $S \gg 1$ the number of species read [15–17]

$$\dot{N_i} = N_i \left( 1 - N_i - \sum_j \alpha_{ij} N_j \right) + \lambda \,. \tag{1}$$

The matrix $\alpha_{ij}$ quantifies the interactions between species, and $\lambda$ accounts for migration of individuals into the community from its surroundings. We consider randomly sampled interaction matrices with independent and identically-distributed entries, such that $\text{mean}(\alpha_{ij}) = \mu/S$ and $\text{std}(\alpha_{ij}) = \sigma/\sqrt{S}$. As the parameter $\sigma$ is increased and crosses a critical value $\sigma_c$, the system exhibits a transition between a fixed point phase and a fluctuating one [3,18], see Fig. 1. The biological relevance of such theoretical descriptions was demonstrated experimentally in [19].

Yet, due to the multiplicative nature of population growth ($\dot{N_i}$ grows with $N_i$), ecological models can exhibit unique properties, and since the dynamics of the population sizes $N_i(t)$ is non-Gaussian, analytical predictions remain challenging. In sharp contrast to the time-

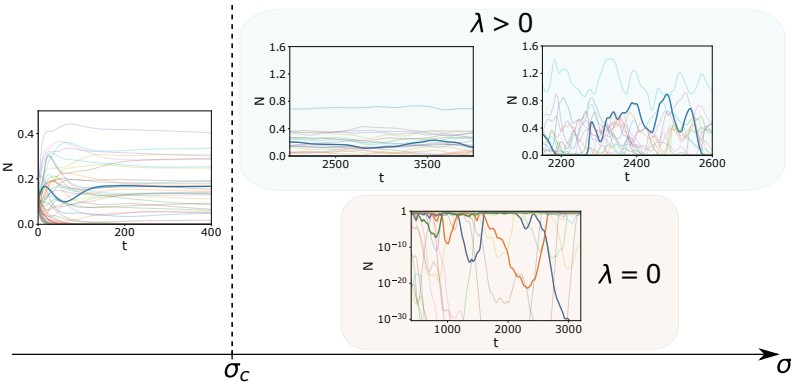

Figure 1: **Phases of the Lotka-Volterra dynamics.** Example degrees of freedom from a simulation of the dynamics Eq. (1) with $\mu = 10$ and $S = 4000$ for increasing values of $\sigma$. When $\sigma < \sigma_c$, the dynamics reach a fixed point. Here $\sigma - \sigma_c = -0.3$. When $\lambda > 0$ and $\sigma > \sigma_c$ the dynamics reach a time-translation invariant state. The amplitude of the time-fluctuations grows with $\sigma$, and their correlation time decreases with $\sigma$. Here $\lambda = 10^{-8}$ and $\sigma - \sigma_c = 0.1$ (Upper left) and $\sigma - \sigma_c = 0.4$ (Upper right). When $\lambda = 0$ and $\sigma > \sigma_c$, the dynamics is aging, with growing fluctuations of log-populations sizes. Here $\sigma - \sigma_c = 0.4$.

translational invariant state reached by other dynamical systems, the dynamics Eq. (1) in the absence of migration (i.e. when $\lambda = 0$) exhibit *aging*. Namely, the correlation time increases with the elapsed time. This was shown numerically in [20], and described analytically in [21], where it was proven that the correlation time increases linearly with the elapsed time. This aging behavior is characterized by variables performing ever larger excursions to values near $N_i = 0$, see Fig. 1, so that the system spends long times in the vicinity of unstable fixed points. This aging is very different from glassy dynamics on a rough landscape [22]. In the high-dimensional limit $S \to \infty$ and at long times, population sizes follow non-Markovian jump-diffusion processes [21]. At positive migration rate $\lambda > 0$, the dynamics do reach a time-translation invariant state [20], characterized by a correlation time that grows as $|\ln \lambda|$ when $\lambda$ is small [21]. The origin of the long timescale can be traced back to population sizes growing and declining exponentially between $O(\lambda)$ and $O(1)$ values. Figure 1 recapitulates these different phases of the Lotka-Volterra dynamics.

Understanding the behavior of these dynamics in the critical regime close to the transition, and how they are affected by the phase space boundaries at $N_i = 0$, has so far remained an open problem. Earlier numerical work showed that, as the transition is approached, the amplitude of the fluctuations gets smaller, and timescales grow, but the precise scaling behavior has not been elucidated [20].

In this work, we provide a comprehensive analytical description of the critical regime of the Lotka-Volterra dynamics Eq. (1) when $\sigma$ goes to $\sigma_c$ from above, meaning from inside the fluctuating phase. We show that there exist three universality classes depending on the relative size of the migration rate $\lambda$, and the distance to the transition $\sigma - \sigma_c$, see Fig. 2(a). One scaling regime is obtained when $\lambda = 0$, where the transition point $\sigma = \sigma_c$ separates a fixed point phase to an aging phase. Another scaling regime is obtained for $\lambda > 0$ fixed, where the transition is from the fixed point phase to a time-translation invariant chaotic phase. This regime is characterized by the same critical exponents as random neural network models with a strongly non-linear transfer function [13]. A third scaling regime is obtained when both $\lambda \to 0^+$ and $\sigma - \sigma_c \to 0^+$, while keeping $\sigma - \sigma_c \gg \sqrt{\lambda}$. For each regime, we describe the dynamics near criticality in terms of a scaling theory for the growth of fluctuations and timescales, and derive the corresponding critical exponents. We find the position of the critical

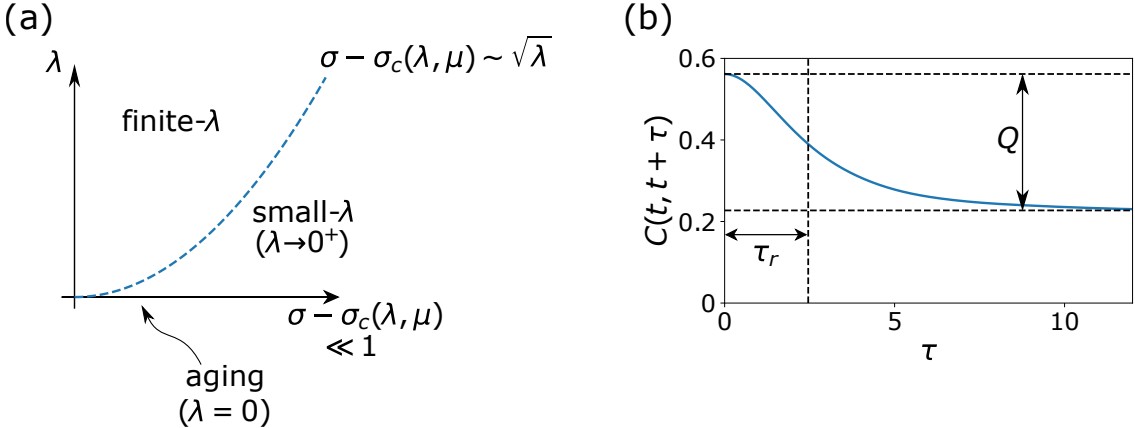

Figure 2: **Critical regimes of the Lotka-Volterra dynamics.** **(a)** Three different scaling regimes are identified when $\sigma - \sigma_c$ is small: when $\lambda = 0$; when $\lambda > 0$ and fixed; and in the limit $\lambda \to 0^+$, taken before the limit $\sigma \to \sigma_c^+$. The crossover between the second and third regimes takes place at $\sigma - \sigma_c \sim \sqrt{\lambda}$. **(b)** Two-time autocorrelation function $C(t, t') \equiv \sum_i N_i(t)N_i(t')/S$ in the fluctuating phase, with definitions of the order parameters $Q$ (the amplitude of the fluctuations) and $\tau_r$ (the relaxation time of the fluctuations). As the critical point is approached from above, $\sigma \to \sigma_c^+$, $Q$ decreases and $\tau_r$ increases. Here $C(t, t')$ is shown for $\lambda = 0.1$, $\mu = 10$ and $\sigma - \sigma_c = 0.1$.

point $\sigma_c(\lambda, \mu)$ and show that the chaotic phase does not exist for large values of $\lambda$.

The paper is organized as follows. We present a summary of our main results in Sec. 2. The rest of the paper is dedicated to the derivation of these results. In Sec. 3, we recall the Dynamical Mean Field Theory (DMFT) equations associated to the Lotka-Volterra dynamics, which form the basis of our analysis. In Sec. 4, we discuss the two cases $\lambda = 0$ and $\lambda \to 0^+$. Then, in Sec. 5, we consider the case where the migration rate $\lambda > 0$ is fixed and positive.

## 2 Summary of the main results

In this section, we summarize the main results for each of the universality classes for the main two quantities of interest: The amplitude of the fluctuations $Q$ and the associated relaxation timescale $\tau_r$, see Fig. 2 (b).

### 2.1 Growth of fluctuations

For fixed parameters $\lambda \geq 0$ and $\mu$, the system is in a fluctuating phase when $\sigma > \sigma_c(\lambda, \mu)$, with $\sigma_c(\lambda, \mu) \to \sqrt{2}$ as $\lambda \to 0$. We study the autocorrelation function of the population sizes $C(t, t') \equiv \sum_i N_i(t)N_i(t')/S$, in the large $S$ limit and for long times $t, t'$, from which we extract the amplitude and relaxation time of the fluctuations, see Fig. 2 (b). The autocorrelation function fully characterizes the long-time dynamics of the population sizes, as explained in Sec. 3. The amplitude $Q$ of the fluctuations is defined from $C(t, t')$ through

$$Q = \lim_{t \to \infty} \left( C(t, t) - \lim_{\tau \to \infty} C(t, t + \tau) \right), \tag{2}$$

which grows continuously from 0 at the transition, with an exponent $\beta$ defined by

$$Q \sim Q_c(\lambda, \mu)|\sigma - \sigma_c(\lambda, \mu)|^\beta. \tag{3}$$

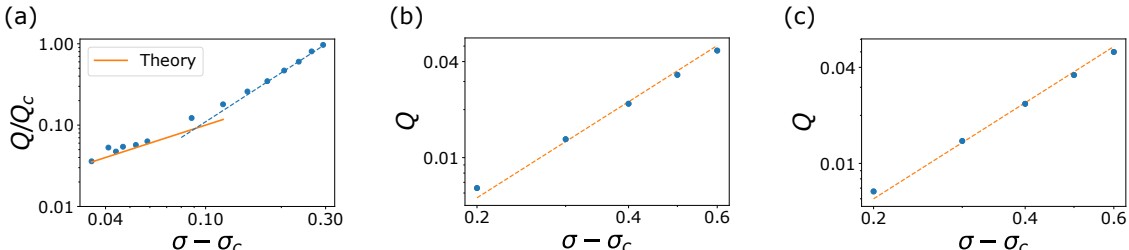

Figure 3: **Growth of fluctuations in the vicinity of the critical point. (a)** $\lambda > 0$ criticality. For small $\sigma - \sigma_c(\lambda, \mu)$ the amplitude of the fluctuations is predicted to grow to leading order as $Q/Q_c = (\sigma - \sigma_c)$ with $Q_c$ a known function of $\lambda$ and $\mu$. The dots are obtained from numerically solving the DMFT equations of Sec. 3. The solid line is the analytical prediction $Q/Q_c = (\sigma - \sigma_c)$. This is a parameter-free agreement, as $Q_c$ is here known analytically. At larger values of $\sigma - \sigma_c(\lambda, \mu)$ the scaling becomes quadratic, as shown by a fit of the form $Q/Q_c = a(\sigma - \sigma_c)^2$ (dashed line). Here $\lambda = 0.1$ and $\mu = 10$. **(b)** $\lambda = 0$ criticality. For small $\sigma - \sigma_c(\lambda = 0, \mu)$ the amplitude of the fluctuations is predicted to grow as $Q \sim (\sigma - \sigma_c)^2$. The dots are obtained from numerically solving the rescaled DMFT equations of Sec. 4.3 and the dashed line is a fit of parameter $Q_c$ to the form $Q = Q_c(\sigma - \sigma_c)^2$. Here $\mu = 10$. **(c)** $\lambda \to 0^+$ criticality. For small $\sigma - \sigma_c(\lambda = 0, \mu)$ the amplitude of the fluctuations is predicted to grow as $Q \sim (\sigma - \sigma_c)^2$. The dots are obtained from numerically solving the rescaled DMFT equations of Sec. 4.1 and the dashed line is a fit of parameter $Q_c$ to the functional form $Q = Q_c(\sigma - \sigma_c)^2$. Here $\mu = 10$.

We obtain the following results in the three regimes of interest. First, when the limit $\lambda \to 0^+$ is taken before the limit $\sigma \to \sigma_c(0, \mu) = \sqrt{2}$ and when $\lambda = 0$, we obtain $\beta = 2$, see Sec. 4.2 and Sec. 4.4. However when $\lambda > 0$ is fixed, we obtain $\beta = 1$, see Sec. 5, Eq. (70). We also find an analytical expression for the coefficient $Q_c(\lambda, \mu)$ in that case, see Sec. 5.3.3. We confirm these predictions in numerical solutions of the DMFT equations established in Sec. 3, see Fig. 3.

## 2.2 Critical slowing down

At long times, the autocorrelation function $C(t, t')$ can be written as a sum of a steady part and a transient part $\delta C(t, t')$,

$$C(t, t') = w^2 + Q \, \delta C(t, t') \tag{4}$$

with $\delta C(t, t) = 1$ and $\lim_{\tau \to \infty} \delta C(t, t + \tau) = 0$. If the limit $\lambda \to 0^+$ is taken at fixed $\sigma > \sqrt{2}$, the dynamics reach a time-translation invariant state with a long correlation time proportional to $|\ln \lambda|$, meaning that the dynamics become regular as $\lambda \to 0^+$ when described over timescales of order $O(|\ln \lambda|)$ [21]. When $|\sigma - \sqrt{2}| \ll 1$, with still $\sigma - \sqrt{2} \gg \sqrt{\lambda}$, the dynamics are described by a critical regime described by the scaling form

$$\delta C(t, t') \sim \delta \hat{C}_+ \left( \frac{|t - t'|/|\ln \lambda|}{|\sigma - \sqrt{2}|^{-\zeta}} \right), \tag{5}$$

with $\zeta = 1$ and $\delta \hat{C}_+$ a scaling function, see Sec. 4.2.

When $\lambda = 0$ the dynamics in the fluctuating phase exhibits an aging behavior with a correlation time that grows linearly with the elapsed time [21]. In other words, the dynamics is time-translation invariant in log-time ($\ln t$). We show that in log-time there is no critical

slowing down, meaning that close to the transition

$$\delta C(t, t') \sim \delta \hat{C}_0 \left( \frac{|\ln t - \ln t'|}{|\sigma - \sqrt{2}|^{-\zeta}} \right), \tag{6}$$

with $\zeta = 0$ and $\delta \hat{C}_0$ another scaling function, see Sec. 4.4.

Lastly, when $\sigma > \sigma_c(\lambda, \mu)$ and $\lambda > 0$ fixed, the dynamics reach a time-translation invariant state. When approaching the critical point $\sigma \to \sigma_c(\lambda, \mu)$ from above while keeping $\lambda$ fixed, the relaxation time of the fluctuations grows with an exponent $\zeta$ inferred from the scaling form

$$\delta C(t, t') \sim 1 - \text{Tanh}^2 \left( \frac{|t - t'|}{\tau_c(\lambda, \mu)|\sigma - \sigma_c(\lambda, \mu)|^{-\zeta}} \right). \tag{7}$$

We show that $\zeta = 1/2$, see the discussion after Eq. (70) in Sec. 5. The scaling function, as well as the value of the parameter $\tau_c(\lambda, \mu)$, are obtained in Eqs. (67) and (74) respectively. We confirm these predictions in numerical solutions of the Dynamical Mean Field Theory equations established in Sec. 3, see Fig. 4.

Note that the correlation functions $\delta \hat{C}_+(x)$ and $\delta \hat{C}_0(x)$ are non-differentiable at $x = 0$, namely the slope $d\left(\delta \hat{C}_+\right)/dx$ has different values at $x \to 0^+$ and $x \to 0^-$. This means that trajectories in rescaled time have a Brownian motion component [21]. In contrast, at $\lambda > 0$, $\delta C(t, t')$ is differentiable at $t = t'$, meaning that trajectories do not have a Brownian motion component.

### 2.3  Crossover between the finite $\lambda > 0$ and the $\lambda \to 0^+$ critical behaviors

For $\lambda \ll 1$, the shift in the critical point is proportional to leading order to $\sqrt{\lambda}$, $\sigma_c(\lambda, \mu) - \sqrt{2} \sim \sqrt{\lambda}$. The amplitude $Q_c(\lambda, \mu)$, see Eq. (3), setting the amplitude of the critical fluctuations beyond the exponent $\beta$, also goes to 0 as

$$Q_c(\lambda, \mu) \sim \sqrt{\lambda},$$

see Eq. (80) in Sec. 5.5. At the same time, the timescale $\tau_c(\lambda, \mu)$ in Eq. (7) increases with

$$\tau_c(\lambda, \mu) \sim \lambda^{-1/4}$$

see Eq. (81) in Sec. 5.5. Combining this with Eqs. (5) and recalling that the exponent in Eq. (3) is $\beta = 2$ when $\lambda \to 0^+$, one finds a crossover at $\sigma - \sqrt{2} \sim \sqrt{\lambda}$ when both $\lambda \ll 1$ and $\sigma - \sqrt{2} \ll 1$. The finite $\lambda > 0$ critical regime dominates for $\sigma - \sqrt{2} \ll \sqrt{\lambda}$ while the $\lambda \to 0^+$ critical regime dominates when $\sigma - \sqrt{2} \gg \sqrt{\lambda}$. The crossover is illustrated in Fig. 3 (a).

## 3  Dynamical mean-field theory

Our theory is built on dynamical mean-field theory (DMFT). Originally developed in the context of spin-glasses [23], and later adapted to ecological dynamics in [2] and to the present equations in [18, 20], it shows that in the limit $S \to \infty$ and for $N_i$ sampled independently at the initial time, the dynamics of the population sizes are described by independent realizations of the stochastic differential equation

$$\dot{N}(t) = N(t)[1 - N(t) - \mu m(t) + \sigma \xi(t)] + \lambda, \tag{8}$$

where the index $i$ has been dropped, and where $\xi(t)$ is a zero-mean Gaussian process and $m(t)$ a deterministic function of time. This can be shown to come from the fact that the term $\xi_i(t) \equiv \sum_j \alpha_{ij} N_j(t)$ in Eq. (1) is the sum of many weakly-correlated contributions. However,

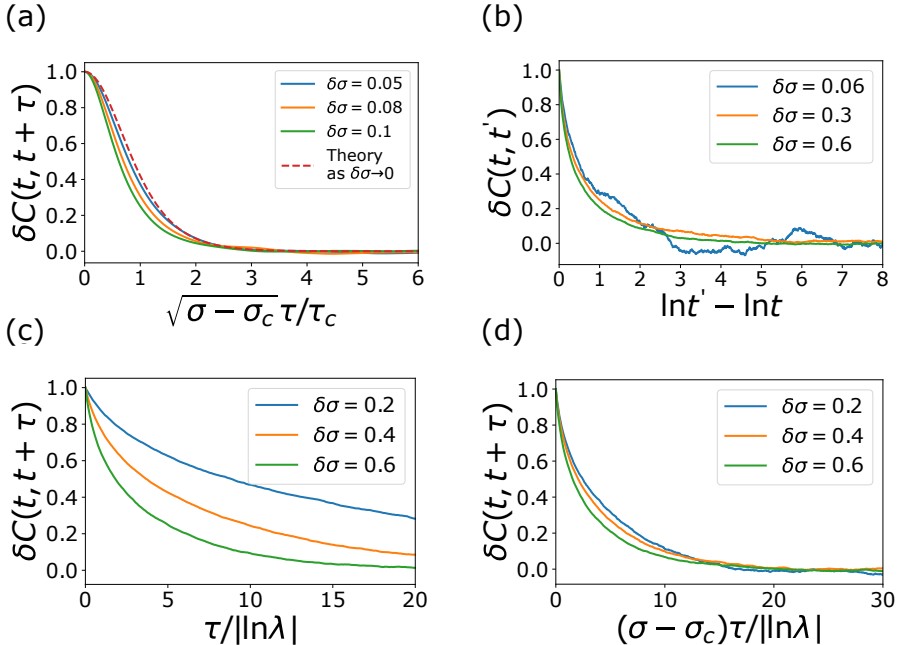

Figure 4: **Growth of timescales in the vicinity of the critical point. (a)** $\lambda > 0$ criticality. For small $\delta\sigma = \sigma - \sigma_c(\lambda,\mu)$, the rescaled correlation function $\delta C(t, t+\tau)$ converges to the scaling form given in Eq. (7). The continuous lines are obtained by numerically solving the DMFT equation of Sec. 3. This is a parameter-free agreement: the dashed red line is the analytical prediction of Eq. (7), and $\tau_c(\lambda,\mu)$ is known. Here $\lambda = 0.1$ and $\mu = 10$. **(b)** $\lambda = 0$ criticality. For small $\delta\sigma = \sigma - \sqrt{2}$, the rescaled correlation function converges to the scaling form of Eq. (6). Note the collapse of the correlation functions for different values of $\delta\sigma$ without rescaling of the log-time axis. The continuous lines are obtained by numerically solving the DMFT equation of Sec. 4.3. Here $\mu = 10$. **(c, d)** $\lambda \to 0^+$ criticality. **(c)** The correlation time of the fluctuations diverge as $\sigma \to \sqrt{2}$ from above. **(d)** For small $\delta\sigma = \sigma - \sqrt{2}$, the timescale in rescaled time grows as $(\sigma - \sqrt{2})^{-1}$. The continuous lines are obtained by numerically solving the DMFT equation of Sec. 4.1. Here $\mu = 10$.

$m(t)$ and the correlations of $\xi(t)$ are not provided in advance. Instead, they are derived through self-consistency conditions. This is a dynamical equivalent of the self-consistency condition on the magnetization derived in the mean-field Ising model, for instance. Recalling that $\langle \alpha_{ij} \rangle = \mu/S$ and $\langle \alpha_{ij}\alpha_{kl} \rangle - \langle \alpha_{ij} \rangle\langle \alpha_{kl} \rangle = \sigma^2 \delta_{ik}\delta_{kl}/S$, the self-consistency conditions read

$$m(t) = \langle N(t) \rangle\,, \tag{9}$$

and

$$\left\langle \xi(t)\xi(t') \right\rangle = \left\langle N(t)N(t') \right\rangle\,, \tag{10}$$

where the average $\langle \dots \rangle$ stands for an average over the stochastic process in (8). Because $\xi(t)$ is a Gaussian process, obtaining the first two moments of the population size $\langle N(t) \rangle$ and $\left\langle N(t)N(t') \right\rangle$ allows to completely characterize the dynamics of $N(t)$ by using Eq. (8).

We denote by $m_\infty$ the steady-state value of $m(t)$. At long-times, we further decompose the noise $\xi(t)$ into a frozen Gaussian random variable $\xi_\infty$ and a Gaussian process $\delta\xi(t)$ that completely decorrelates over time, meaning $\xi(t) \equiv \xi_\infty + \epsilon\delta\xi(t)$ with $\lim_{\tau\to\infty} \langle \delta\xi(t)\delta\xi(t+\tau) \rangle = 0$. The amplitude $\epsilon$ of the fluctuating part is defined so that $\left\langle \delta\xi(t)^2 \right\rangle = 1$. Using the self-consistency condition Eq. (10), we get from the decomposition in Eq. (4) that $\left\langle \xi_\infty^2 \right\rangle = w^2$ and $\langle \delta\xi(t)\delta\xi(t+\tau) \rangle = \delta C(t, t+\tau)$. We also obtain $Q = \epsilon^2$. To make the notations more compact, we introduce $g \equiv (1-\mu m_\infty)/\sigma + \xi_\infty$ and $\tilde{m} \equiv (1-\mu m_\infty)/\sigma$. The dynamics in Eq. (8) then becomes at long-times

$$\dot{N}(t) = N(t)[\sigma(g + \epsilon\delta\xi(t)) - N(t)] + \lambda\,. \tag{11}$$

The random variable $g$ is Gaussian distributed with yet unknown mean and variance. Its distribution is denoted $P(g)$ and is given by

$$P(g) = \frac{1}{\sqrt{2\pi}w} \exp\left(-\frac{(g-\tilde{m})^2}{2w^2}\right)\,. \tag{12}$$

We recall that the average $\langle \dots \rangle$ runs over realizations of the frozen variable $g$ and the Gaussian process $\delta\xi(t)$. Below it will be convenient to perform this average in two parts. We thus introduce the notation $\langle \dots \rangle_g$ to denote an average over the Gaussian process $\delta\xi(t)$ at fixed $g$. Equations (9,10,11,12) form the basis of the subsequent analysis.

# 4 Critical regimes when $\lambda \to 0^+$ and $\lambda = 0$

We now determine the properties of the dynamical phase transition in the two cases where the limit $\lambda \to 0^+$ is taken before the limit $\sigma \to \sqrt{2}$ and where $\lambda = 0$. As discussed in Sec. 2.3, this first regime is relevant when $\lambda \ll 1$, $\sigma - \sqrt{2} \ll 1$ and $\lambda \ll \sigma - \sqrt{2}$. In those two cases, for any $\sigma > \sqrt{2}$ finite, it was shown in [21], that the dynamics evolve over infinitely long timescales.

When $\lambda \to 0^+$, the dynamics is time-translation invariant at long times with a long correlation time proportional to $|\ln\lambda|$, with

$$\lim_{\lambda\to0^+} \delta C_\lambda(t, t+|\ln\lambda|s) \to \delta\hat{C}_+(s)\,,$$

with $\delta\hat{C}_+(s)$ a regular function as $s \to 0^+$.

In the absence of migration ($\lambda = 0$),

$$\lim_{t\to\infty} \delta C_{\lambda=0}(t, te^s) = \delta\hat{C}_0(s)\,,$$

with $\delta\hat{C}_0(s)$ another regular function as $s \to 0^+$. This is typical of an aging dynamics, with a growth of timescale proportional to the age of the system. Leveraging on these two identities,

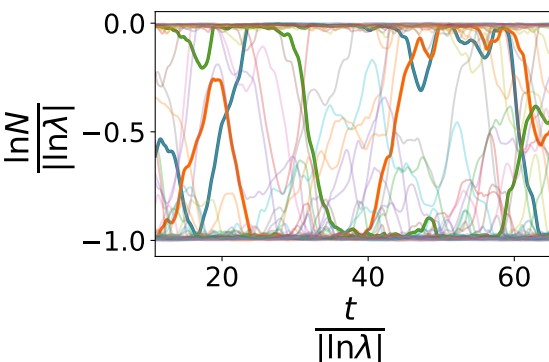

Figure 5: **Slow dynamics in the fluctuating phase for $\lambda \ll 1$.** Example degrees of freedom from a simulation of the dynamics Eq. (1) with $\lambda = 10^{-40}$, $\mu = 10$, $\sigma - \sigma_c = 0.4$ and $S = 4000$. The process $z(s)$ with $s = t/|\ln \lambda|$ and $z = \ln N/|\ln \lambda|$ converges when $\lambda \to 0^+$ to a time-translation invariant process, confined between 0 and $-1$, with finite correlation time.

effective equations for the long-time dynamics of the population size $N(t)$ were obtained in [21]. These long-time effective dynamics are reviewed in Sec. 4.1 and Sec. 4.3, for dynamics with $\lambda \to 0^+$ and $\lambda = 0$ respectively. We then use these results to derive the corresponding scaling behaviors, in Sec. 4.2 and Sec. 4.4.

## 4.1 Steady-state dynamics when $\lambda \to 0^+$

When $\lambda \to 0^+$ and $\sigma - \sqrt{2} > 0$ fixed, the effective steady-state dynamics is described by a rescaling of the original DMFT equations Eq. (11), as illustrated in Fig. 5. We introduce $z = \ln N/|\ln \lambda|$ and $s = t/|\ln \lambda|$. When $\lambda \to 0^+$, the process $z(s)$ follows a well-defined stochastic differential equation [21],

$$z'(s) = \sigma g + \sigma \epsilon \delta \hat{\xi}(s) - W(z) + W(-z-1) \tag{13}$$

with $\delta \hat{\xi}(s) \equiv \delta \xi(t)$ a zero-mean Gaussian noise with correlations $\delta \hat{C}_+(s)$. The function $W(z)$ acts as a hard wall with $W(z > 0) = +\infty$ and $W(z < 0) = 0$, so that the dynamics are confined between $-1 \leq z \leq 0$. Equation (13) is supplemented by an expression for $N(s)$ valid at long times,

$$N(s) = \sigma \Theta(z(s))\left(g + \epsilon \delta \hat{\xi}(s)\right), \tag{14}$$

where the Heaviside function $\Theta$ is used with the convention $\Theta(0) = 1$. Therefore, the system of self-consistency equations in the $\lambda \to 0^+$ regime reads

$$\frac{1 - \sigma \tilde{m}}{\mu} = \sigma \left\langle \Theta(z(s))\left(g + \epsilon \delta \hat{\xi}(s)\right) \right\rangle, \tag{15}$$

and

$$w^2 + \epsilon^2 \delta \hat{C}_+(s) = \sigma^2 \left\langle \Theta(z(s))\Theta(z(0))\left(g + \epsilon \delta \hat{\xi}(s)\right)\left(g + \epsilon \delta \hat{\xi}(0)\right) \right\rangle, \tag{16}$$

where all averages are performed in steady-state.

## 4.2 Growth of fluctuations and timescales when $\lambda \to 0^+$

Equation (16), together with the dynamics in Eq. (13), gives us the self-consistency equation satisfied by the correlation function $\delta \hat{C}_+(s)$. In the right-hand side of Eq. (16), we split the average over $g$ between (i) $g > 0$, in which case $\Theta(z) = 1$ with high probability for small $\epsilon$,

and (ii) $g < 0$ in which case $\Theta(z) = 0$ with high probability for small $\epsilon$. We thus rewrite Eq. (16) as

$$
w^2 + \epsilon^2 \delta \hat{C}_+(s) = \sigma^2 \int_0^{+\infty} \mathrm{d}g \, P(g) \left( g^2 + \epsilon^2 \delta C_+(s) \right)
$$
$$
+ \sigma^2 \int_{-\infty}^{+\infty} \mathrm{d}g \, P(g) \left\langle \left[ \Theta(z(s)) \Theta(z(0)) - \Theta(g) \right] (g + \epsilon \, \delta \xi(0)) (g + \epsilon \, \delta \xi(s)) \right\rangle_g .
$$
(17)

No approximation is made in this rewriting. The first integral on the right-hand side of the above equation can be computed using the definition of $P(g)$ in Eq. (12), so the above equation becomes

$$
\left( 1 - \frac{\sigma^2}{2} \right) \left( w^2 + \epsilon^2 \delta \hat{C}_+(s) \right) + \frac{\sigma^2}{2} \left( w^2 + \epsilon^2 \delta \hat{C}_+(s) \right) \mathrm{Erf}\left( \frac{\tilde{m}}{\sqrt{2}w} \right)
$$
$$
+ \frac{\sigma^2}{2} \tilde{m}^2 \left[ 1 + \mathrm{Erf}\left( \frac{\tilde{m}}{\sqrt{2}w} \right) \right] + \frac{\tilde{m}w}{\sqrt{2\pi}} \exp\left( -\frac{\tilde{m}^2}{2w^2} \right) = \epsilon^3 P(g = 0) I(s),
$$

where we introduced

$$
I(s) \equiv \frac{\sigma^2}{\epsilon^3 P(g = 0)} \int_{-\infty}^{+\infty} \mathrm{d}g \, P(g) \left\langle \left[ \Theta(z(s)) \Theta(z(0)) - \Theta(g) \right] (g + \epsilon \, \delta \xi(0)) (g + \epsilon \, \delta \xi(s)) \right\rangle_g .
$$

We recall that $\delta \hat{C}_+(\infty) = 0$, so that

$$
\left( 1 - \frac{\sigma^2}{2} \right) w^2 + \frac{\sigma^2}{2} w^2 \mathrm{Erf}\left( \frac{\tilde{m}}{\sqrt{2}w} \right) + \frac{\sigma^2}{2} \tilde{m}^2 \left[ 1 + \mathrm{Erf}\left( \frac{\tilde{m}}{\sqrt{2}w} \right) \right] + \frac{\tilde{m}w}{\sqrt{2\pi}} \exp\left( -\frac{\tilde{m}^2}{2w^2} \right) = \epsilon^3 P(g = 0) I(\infty).
$$
(18)

Furthermore, $\delta \hat{C}_+(0) = 1$ so that we get

$$
\epsilon^{-1} \left( 1 - \frac{\sigma^2}{2} + \frac{\sigma^2}{2} \mathrm{Erf}\left( \frac{\tilde{m}}{\sqrt{2}w} \right) \right) = P(g = 0) (I(0) - I(\infty)).
$$
(19)

Hence we obtain an exact nonlinear equation satisfied by the correlation function

$$
\delta \hat{C}_+(s) = \frac{I(s) - I(\infty)}{I(0) - I(\infty)}.
$$
(20)

We can now study this equation when $\epsilon \ll 1$. In that case, the contribution to $I(s)$ coming from $O\left( \epsilon^0 \right)$ values of $g$ are exponentially small in $1/\epsilon$. The only perturbative contributions to $I(s)$ arise from the values $g = O(\epsilon)$. We therefore introduce $\tilde{g} \equiv g/\epsilon$. Accordingly, we rescale time as $\tilde{s} = \epsilon s$ and denote $\tilde{z}(\tilde{s}) \equiv z(\tilde{s}/\epsilon)$ which satisfies an equation like that for the original process $z(s)$, Eq. (13), but with an $O(1)$ fluctuating noise,

$$
\frac{\mathrm{d}\tilde{z}}{\mathrm{d}\tilde{s}} = \sigma (\tilde{g} + \delta \xi(\tilde{s})) - W(\tilde{z}) + W(-\tilde{z} - 1).
$$
(21)

In terms of this process, we get

$$
I(\tilde{s}) = \frac{\sigma^2}{P(g = 0)} \int_{-\infty}^{+\infty} \mathrm{d}\tilde{g} \, P(\epsilon \tilde{g}) \left\langle \left[ \Theta(\tilde{z}(\tilde{s})) \Theta(\tilde{z}(0)) - \Theta(\tilde{g}) \right] (\tilde{g} + \delta \xi(0)) (\tilde{g} + \delta \xi(\tilde{s})) \right\rangle_{\tilde{g}}
$$
(22)

Hence $I = O\left( \epsilon^0 \right)$, and to leading order

$$
I(\tilde{s}) = \sigma^2 \int_{-\infty}^{+\infty} \mathrm{d}\tilde{g} \, \left\langle \left[ \Theta(\tilde{z}(\tilde{s})) \Theta(\tilde{z}(0)) - \Theta(\tilde{g}) \right] (\tilde{g} + \delta \xi(0)) (\tilde{g} + \delta \xi(\tilde{s})) \right\rangle_{\tilde{g}} + O\left( \epsilon \right),
$$
(23)

as $P(\epsilon \tilde{g})$ can be replaced by $P(0)$. A similar reasoning can be applied to the first moment equation, Eq. (15), and yields,

$$\left(\frac{1}{\mu} - \frac{\sigma w}{\sqrt{2\pi}}\right) - \frac{\sigma \tilde{m}}{\mu} - \frac{\sigma w}{\sqrt{2\pi}}\left[\exp\left(-\frac{\tilde{m}^2}{2w^2}\right) - 1\right] - \frac{\sigma \tilde{m}}{2}\left[1 + \mathrm{Erf}\left(\frac{\tilde{m}}{\sqrt{2}w}\right)\right] = \epsilon^2 P(g=0)J, \quad (24)$$

with

$$J = \frac{\sigma}{P(g=0)}\int_{-\infty}^{+\infty} d\tilde{g}\, P(\epsilon \tilde{g})\left\langle[\Theta(\tilde{z}(\tilde{s})) - \Theta(\tilde{g})](\tilde{g} + \delta\xi(\tilde{s}))\right\rangle_{\tilde{g}},$$

$$= \sigma\int_{-\infty}^{+\infty} d\tilde{g}\,\left\langle[\Theta(\tilde{z}(\tilde{s})) - \Theta(\tilde{g})](\tilde{g} + \delta\xi(\tilde{s}))\right\rangle_{\tilde{g}} + O(\epsilon).$$

We can now obtain the scaling behavior close to the critical point. Equation (20) provides a nonlinear equation for the leading order correlation function $\delta\hat{C}_+(\tilde{s})$, when $I(\tilde{s})$ is replaced by its leading order expression in Eq. (23). This equation cannot be solved explicitly since the average entering Eq. (23) cannot be performed. However, both Eq. (23) to leading order, and the dynamical equation (21) do not depend on $\tilde{m}$, $w$ and $\epsilon$. These two equations then fix $\delta\hat{C}_+(\tilde{s})$ to lowest order near the critical point. Given $\delta\hat{C}_+(\tilde{s})$, we are then left with three equations Eqs. (18,19,24) to solve for $\tilde{m}$, $w$ and $\epsilon$ near the transition. At the transition, $\sigma = \sqrt{2}$, $\tilde{m} = 0$, $w = \sqrt{\pi}/\mu$ and $\epsilon = 0$. For $\sigma = \sqrt{2} + x$ with $0 < x \ll 1$, we expand close to the transition

$$\tilde{m} = m_1 x + m_2 x^2 + \dots,$$

and

$$w = \frac{\sqrt{\pi}}{\mu}(1 + w_1 x + \dots),$$

and lastly

$$\epsilon = \epsilon_1 x + \dots \quad (25)$$

The form of the expansion for the amplitude $\epsilon$ of the fluctuations in Eq. (25) is imposed by Eq. (19), recalling that $\sigma = \sqrt{2} + x$. It turns out that the first corrections to $\tilde{m}$ and $w$ can be derived explicitly from Eqs. (18,24) alone. We find that $\tilde{m}$ agrees with the analytical continuation of the fixed point branch [3] up to order $O(x^2)$ with

$$m_1 = -\frac{\pi}{2\mu} \quad \text{and} \quad m_2 = \frac{-4\pi^2 + 10\pi\mu - 3\pi^2\mu}{16\sqrt{2}\mu^2}, \quad (26)$$

while $w$ agrees with the analytical continuation of the fixed point branch up to order $O(x)$ with

$$w_1 = \frac{2\pi - 2\mu + \pi\mu}{2\sqrt{2}\mu}. \quad (27)$$

To lowest order, Eq. (19) gives the expression of the amplitude of fluctuations close to the critical point

$$\epsilon_1 = \frac{\pi}{2\mu(I(\infty) - I(0))}. \quad (28)$$

Since $Q = \epsilon^2$, this equation yields the critical exponent $\beta = 2$ as defined through Eq. (3). The expression for $\epsilon_1$ is not explicit, as it relies of the solution of the nonlinear equation Eq. (20) for the correlation function, so $Q_c(\lambda, \mu)$ in Eq. (3) is not given explicitly.. The rescaling of time used to obtain Eq. (20) entails the scaling exponent $\zeta = \beta/2 = 1$. As we show below, the scaling relation $\zeta = \beta/2$ is common to both the finite $\lambda > 0$ and $\lambda \to 0^+$ critical regimes. The picture behind these scaling results is the following: when the temporal fluctuations in $\xi(t)$ are small, of order $\epsilon$, only species whose time-averaged growth rate, $1 - \mu m + \sigma\overline{\xi_i(t)}$, is

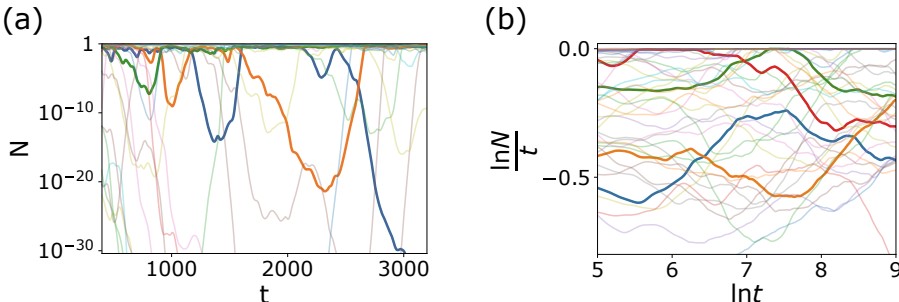

Figure 6: **Aging dynamics and time-translation invariant dynamics of the process $z(s)$.** Example degrees of freedom from a simulation of the dynamics Eq. (1) with $\lambda = 0$, $\mu = 10$, $\sigma - \sigma_c = 0.4$ and $S = 4000$. **(a)** Growth of the relaxation time and amplitude of the fluctuations of log-population sizes with the elapsed time in the aging regime. **(b)** Corresponding time-translation invariant process $z(s)$ where $s = \ln t$ and $z = \ln N / t$.

$O(\epsilon)$ show any significant fluctuations in their $\ln N_i$ values. The instantaneous growth rates of those that do is of order $\epsilon$ (negative or positive), and so the exponential growth and decline of $N_i$ between $O(\lambda)$ and $O(1)$ (see Fig. 5) , takes time of order $|\ln \lambda| / \epsilon$. As, by definition, $Q \sim \epsilon^2$ this explains the scaling relation $\beta = \zeta / 2$.

## 4.3    Steady-state dynamics when $\lambda = 0$

When $\lambda = 0$, the long-time dynamics can be described in a similar way using a Lamperti transformation of the original equations of motion Eq. (11), see Fig. 6. We introduce $z = \ln N / t$ and $s = \ln t$. When $t \to \infty$, the process $z(s)$ follows a well-defined stochastic differential equation [21],

$$z'(s) = -z(s) + \sigma g + \sigma \epsilon \delta \hat{\xi}(s) + W(z), \qquad (29)$$

confined to $z \leq 0$ and with with $\delta \hat{\xi}(s) \equiv \delta \xi(t)$ a zero-mean Gaussian noise with correlations $\delta \hat{C}_0(s)$. The expression for the population size when $\lambda = 0$ remains the same as in Sec. 4.1, with

$$N(s) = \sigma \Theta(z(s)) \big( g + \epsilon \delta \hat{\xi}(s) \big) .$$

Therefore, the self-consistency conditions Eqs. (15,16) also hold upon replacing $\delta \hat{C}_+(s)$ by $\delta \hat{C}_0(s)$. The $\lambda = 0$ and $\lambda \to 0^+$ long-time effective dynamics differ by the nature of the process $z(s)$: when $\lambda = 0$, the process is confined on the negative side by a harmonic potential whereas it is confined by another hard wall at $z = -1$ when $\lambda \to 0^+$. This seemingly innocuous difference entails a profound distinction when it comes to critical slowing down. However, the formal resemblance between the two dynamics allows us to investigate their critical property in a similar way.

## 4.4    Growth of fluctuations when $\lambda = 0$

When $\lambda = 0$, the calculation proceeds in a similar way, with yet a crucial difference. When $\lambda \to 0^+$, we obtained to leading order an $\epsilon$-independent equation for the correlation function by introducing the rescaled process $\tilde{z}(\tilde{s}) \equiv z(\tilde{s} / \epsilon)$. Here one achieves a similar conclusion by rescaling $z$ following $\tilde{z}(s) \equiv z(s) / \epsilon$. From Eq. (29), we find that the latter obeys

$$\frac{d\tilde{z}}{ds} = -z + \sigma \left( \tilde{g} + \delta \xi(s) \right) - W(\tilde{z}) .$$

The rest follows as before and to leading order,

$$\delta \hat{C}_0(s) = \frac{I(\infty) - I(s)}{I(\infty) - I(0)}, \tag{30}$$

with

$$I(s) = \int_{-\infty}^{+\infty} d\tilde{g} \, \langle [\Theta(\tilde{z}(s))\Theta(\tilde{z}(0)) - \Theta(\tilde{g})](\tilde{g} + \delta\xi(0))(\tilde{g} + \delta\xi(s)) \rangle_{\tilde{g}} \, .$$

The expansion around $\sigma = \sqrt{2}$ of $\tilde{m}$, $w$ and $\epsilon$ remains the same. Thus Eqs. (26,27,28) remain valid and the exponent $\beta = 2$ is preserved. The absence of rescaling of time to obtain Eq. (30) entails the scaling exponent $\zeta = 0$, meaning that there is no critical slowing down of the log-time dynamics.

## 5 Finite $\lambda$ criticality

In this section, we investigate the properties of the critical point when the migration rate $\lambda > 0$ is finite. The amplitude of the fluctuations vanishes at the critical point, so that $\epsilon \to 0$. Furthermore, we will show that the correlation-time of $\delta\xi(t)$ diverges at the critical point, see Eq. (7), a form of critical slowing down. Hence, near the transition $\delta\xi(t)$ changes slowly in time. We thus perform an "adiabatic" expansion of $N(t)$, around the fixed point value of Eq. (11) that it would reach if $\delta\xi(t)$ was constant in time, which is

$$\bar{N}(t) = \frac{\sigma\left(g + \epsilon\delta\xi(t)\right) + \sqrt{\sigma^2\left(g + \epsilon\delta\xi(t)\right)^2 + 4\lambda}}{2} \, . \tag{31}$$

We now introduce the deviation $\delta N(t) = N(t) - \bar{N}(t)$. We anticipate that $\delta N(t)$ is small close to the transition and define $\widehat{\delta N}(t) \equiv \delta N(t)/\epsilon$. Differentiating the dynamical mean-field equation Eq. (11), yields exactly

$$\frac{d\widehat{\delta N}(t)}{dt} = -\left(\sqrt{\sigma^2\left(g + \epsilon\delta\xi(t)\right)^2 + 4\lambda} + \epsilon\,\widehat{\delta N}(t)\right)\widehat{\delta N}(t) - \frac{\sigma}{2}\left(1 + \frac{\sigma\left(g + \epsilon\delta\xi(t)\right)}{\sqrt{\sigma^2\left(g + \epsilon\delta\xi(t)\right)^2 + 4\lambda}}\right)\dot{\delta\xi}(t). \tag{32}$$

We can now rewrite the self-consistency conditions Eqs. (9,10) in terms of the processes $\widehat{\delta N}(t)$ and $\bar{N}(t)$, which will be at the basis of our expansion close to the critical point. Lastly, it is known that at long times the dynamics in Eq. (11) reaches a time-translation invariant state [20], so that one can replace $\delta C(t, t + \tau)$ by $\delta C(\tau)$. This leads to the self-consistency equations

$$\frac{1 - \sigma\tilde{m}}{\mu} - \int_{-\infty}^{+\infty} dg\, P(g) \langle \bar{N}(t) \rangle_g = \epsilon \int_{-\infty}^{+\infty} dg\, P(g) \langle \widehat{\delta N}(t) \rangle_g \, , \tag{33}$$

and

$$w^2 + \epsilon^2 \delta C(t) - \int_{-\infty}^{+\infty} dg\, P(g) \langle \bar{N}(t)\bar{N}(0) \rangle_g = \epsilon \int_{-\infty}^{+\infty} dg\, P(g) \langle \bar{N}(0)\widehat{\delta N}(t) \rangle_g$$

$$+ \epsilon \int_{-\infty}^{+\infty} dg\, P(g) \langle \bar{N}(t)\widehat{\delta N}(0) \rangle_g + \epsilon^2 \int_{-\infty}^{+\infty} dg\, P(g) \langle \widehat{\delta N}(0)\widehat{\delta N}(t) \rangle_g \, . \tag{34}$$

We introduce the notation

$$\chi_{\tilde{m},w}\left(\epsilon^2\right) \equiv \int_{-\infty}^{+\infty} dg\, P(g) \langle \bar{N}(t) \rangle_g \, ,$$

which depends on the parameters $\tilde{m}$, $w$ and is regarded as a function of $\epsilon^2$. We also introduce

$$\mathcal{H}_{\tilde{m},w}\left(x = \epsilon^2, y = \epsilon^2 \delta C(t)\right) \equiv \int_{-\infty}^{+\infty} \mathrm{d}g \, P(g) \left\langle \bar{N}(t)\bar{N}(0) \right\rangle_g,$$

which depends on the parameters $\tilde{m}$, $w$ and is regarded as a function of the two variables $x = \epsilon^2$ and $y = \epsilon^2 \delta C(t)$. The fact that $\delta\xi(t)$, appearing in the definition of $\bar{N}(t)$ in Eq. (31), is Gaussian guarantees the existence of this functional form. Lastly, we also introduce

$$\mathcal{I}_{\tilde{m},w}[\delta C](t) \equiv \epsilon^{-1} \int_{-\infty}^{+\infty} \mathrm{d}g \, P(g) \left\langle \bar{N}(0)\widehat{\delta N}(t) \right\rangle_g + \epsilon^{-1} \int_{-\infty}^{+\infty} \mathrm{d}g \, P(g) \left\langle \bar{N}(t)\widehat{\delta N}(0) \right\rangle_g$$
$$+ \int_{-\infty}^{+\infty} \mathrm{d}g \, P(g) \left\langle \widehat{\delta N}(0)\widehat{\delta N}(t) \right\rangle_g, \tag{35}$$

which is a functional of the correlation function. It therefore depends on $\delta C(s)$ for all $0 \le s \le t$. With these notations we can rewrite Eq. (34) in a compact way

$$w^2 + \epsilon^2 \delta C(t) - \mathcal{H}_{\tilde{m},w}\left(\epsilon^2, \epsilon^2 \delta C(t)\right) = \epsilon^2 \mathcal{I}_{\tilde{m},w}[\delta C](t). \tag{36}$$

This allows to obtain a compact equation satisfied by the correlation function $\delta C(t)$ as follows. First, when $t \to \infty$, we have by definition $\delta C(t) \to 0$. Hence

$$w^2 = \mathcal{H}_{\tilde{m},w}\left(\epsilon^2, 0\right) + \epsilon^2 \mathcal{I}_{\tilde{m},w}[\delta C](\infty). \tag{37}$$

Therefore,

$$\epsilon^2 \delta C(t) - \left(\mathcal{H}_{\tilde{m},w}\left(\epsilon^2, \epsilon^2 \delta C(t)\right) - \mathcal{H}_{\tilde{m},w}\left(\epsilon^2, 0\right)\right) = \epsilon^2 \left(\mathcal{I}_{\tilde{m},w}[\delta C](t) - \mathcal{I}_{\tilde{m},w}[\delta C](\infty)\right),$$

which, in the chaotic phase where $\epsilon > 0$, simplifies to

$$\delta C(t) - \frac{\mathcal{H}_{\tilde{m},w}\left(\epsilon^2, \epsilon^2 \delta C(t)\right) - \mathcal{H}_{\tilde{m},w}\left(\epsilon^2, 0\right)}{\epsilon^2} = \mathcal{I}_{\tilde{m},w}[\delta C](t) - \mathcal{I}_{\tilde{m},w}[\delta C](\infty). \tag{38}$$

So far, all these self-consistency equations are exact. In the following, we solve them in the vicinity of the critical point. We start by showing that the solution $\delta C(t)$ of Eq. (38) exhibits a diverging correlation time when $\epsilon \to 0$. We then use this fact to obtain the critical value $\sigma_c(\mu, \lambda)$ above which the system enters the chaotic phase. This generalizes earlier results ( [3]) showing that $\sigma_c(\mu, \lambda) \to \sqrt{2}$ as $\lambda \to 0^+$. We also obtain predictions for $\tilde{m}_c$ and $w_c$, the values of $\tilde{m}$ and $w$ respectively, at the transition. Then we investigate the vicinity of the critical point, get the critical exponents and derive the solution $\delta C(t)$ of Eq. (38) to leading order when $\epsilon \to 0$. Lastly, we investigate the fate of these results when $\lambda \ll 1$.

## 5.1 Existence of critical slowing down

We start by assuming that there is no critical slowing down and that the timescales over which $\delta C(t)$ evolves remain finite when approaching the transition from above. We will eventually show that this ansatz does not lead to any physical solution of Eq. (38). Let $\delta C_0(t) \equiv \lim_{\sigma \to \sigma_c} \delta C(t)$ be the correlation function to zeroth order in $\epsilon$. An equation for $\delta C_0(t)$ can then be obtained from Eq. (38) as follows. As $\epsilon \to 0$ (or equivalently $\sigma \to \sigma_c$), we get

$$\delta C_0(t)\left(1 - \partial_y \mathcal{H}_{\tilde{m}_c,w_c}(0,0)\right) = \lim_{\epsilon \to 0} \left(\mathcal{I}_{\tilde{m},w}[\delta C](t) - \mathcal{I}_{\tilde{m},w}[\delta C](\infty)\right). \tag{39}$$

We denote by $P_c(g)$ the probability distribution of $g$, Eq. (12), at the transition where $\tilde{m}$ and $w$ are replaced by their value at the critical point, $\tilde{m}_c$ and $w_c$ respectively. By definition, we have

$$\mathcal{I}_{\tilde{m},w}[\delta C](\tau) - \mathcal{I}_{\tilde{m},w}[\delta C](\infty) = \epsilon^{-1} \int_{-\infty}^{+\infty} dg\, P(g) \left[ \left\langle \bar{N}(0)\widehat{\delta N}(\tau) + \bar{N}(\tau)\widehat{\delta N}(0) \right\rangle_g - 2 \left\langle \bar{N} \right\rangle_g \left\langle \widehat{\delta N} \right\rangle_g \right]$$
$$+ \int_{-\infty}^{+\infty} dg\, P(g) \left[ \left\langle \widehat{\delta N}(0)\widehat{\delta N}(\tau) \right\rangle_g - \left\langle \widehat{\delta N} \right\rangle_g^2 \right].$$

We further introduce the functions

$$f(x) = \frac{x + \sqrt{x^2 + 4\lambda}}{2}, \tag{40}$$

so that $\bar{N}(t) = f(\sigma(g + \epsilon\delta\xi(t)))$ and

$$h(x) = \sqrt{x^2 + 4\lambda},$$

so that

$$\frac{d\widehat{\delta N}(t)}{dt} = -\left( h(\sigma(g + \epsilon\delta\xi(t))) + \epsilon\,\widehat{\delta N}(t) \right) \widehat{\delta N}(t) - \sigma f'(\sigma(g + \epsilon\delta\xi(t)))\,\dot{\delta\xi}(t).$$

We can expand the value of $\bar{N}(t)$ around $f(g)$ to get to leading order

$$\epsilon^{-1} \int_{-\infty}^{+\infty} dg\, P(g) \left[ \left\langle \bar{N}(0)\widehat{\delta N}(\tau) + \bar{N}(\tau)\widehat{\delta N}(0) \right\rangle_g - 2 \left\langle \bar{N} \right\rangle_g \left\langle \widehat{\delta N} \right\rangle_g \right]$$
$$= \sigma \int_{-\infty}^{+\infty} dg\, P(g) f'(\sigma g) \left[ \left\langle \delta\xi(0)\widehat{\delta N}(\tau) \right\rangle_g + \left\langle \delta\xi(\tau)\widehat{\delta N}(0) \right\rangle_g \right] + O(\epsilon).$$

Hence we obtain

$$\lim_{\epsilon \to 0} \left( \mathcal{I}_{\tilde{m},w}[\delta C](t) - \mathcal{I}_{\tilde{m},w}[\delta C](\infty) \right) = \sigma \int_{-\infty}^{+\infty} dg\, P_c(g) f'(\sigma g) \left[ \left\langle \delta\xi(0)\widehat{\delta N}(\tau) \right\rangle_g + \left\langle \delta\xi(\tau)\widehat{\delta N}(0) \right\rangle_g \right]$$
$$+ \int_{-\infty}^{+\infty} dg\, P_c(g) \left\langle \widehat{\delta N}(0)\widehat{\delta N}(t) \right\rangle_g.$$

Therefore, to get the right-hand side of Eq. (39), we can replace $\delta\xi(t)$ by a zero-mean Gaussian process $\delta\xi_0(t)$ with correlations $\langle \delta\xi_0(t)\delta\xi_0(0) \rangle = \delta C_0(t)$ and we can also approximate $\widehat{\delta N}(t)$ by the process $\widehat{\delta N}_0(t)$, which obeys

$$\frac{d\widehat{\delta N}_0(t)}{dt} = -h(\sigma_c g)\widehat{\delta N}_0(t) - \sigma_c f'(\sigma_c g)\,\dot{\delta\xi}_0(t), \tag{41}$$

and which is solved at a steady-state by

$$\widehat{\delta N}_0(t) = -\sigma_c f'(\sigma_c g) \int_{-\infty}^{t} du\, \exp(-(t-u)h(\sigma_c g))\,\dot{\delta\xi}_0(u).$$

We can now write

$$\left\langle \widehat{\delta N}_0(t)\widehat{\delta N}_0(0) \right\rangle_g = -\sigma_c^2 f'(\sigma_c g)^2 \int_{-\infty}^{+\infty} du\, \frac{\exp(-|u|h(\sigma_c g))}{2h(\sigma_c g)} \delta C_0''(t-u), \tag{42}$$

together with

$$\left\langle \delta\xi_0(0)\widehat{\delta N}_0(t) + \delta\xi_0(t)\widehat{\delta N}_0(0)\right\rangle_g = -\sigma_c f'(\sigma_c g)\int_{-\infty}^{+\infty} du \exp\left(-|u|h(\sigma_c g)\right)(2\Theta(u)-1)\,\delta C_0'(t-u).$$
(43)

To leading order in the vicinity of the critical point, the correlation function $\delta C_0(t)$ therefore satisfies the integro-differential equation

$$\left(1 - \partial_y \mathcal{H}_{\tilde{m}_c, w_c}(0,0)\right)\delta C_0(t) = \int_{-\infty}^{+\infty} du\, K_1(u)\,\delta C_0''(t-u) + \int_{-\infty}^{+\infty} du\, K_2(u),\delta C_0'(t-u).$$

The kernels follow from Eqs. (42,43) and read

$$K_1(u) = -\int_{-\infty}^{+\infty} dg\, P_c(g)\left[\sigma_c^2 f'(\sigma_c g)^2 \frac{\exp\left(-|u|h(\sigma_c g)\right)}{2h(\sigma_c g)}\right],$$

and

$$K_2(u) = -\int_{-\infty}^{+\infty} dg\, P_c(g)\left[\sigma_c^2 f'(g)^2 \exp\left(-|u|h(\sigma_c g)\right)(2\Theta(u)-1)\right].$$

Noting that $K_2(u) = -2K_1'(u)$, integrating by parts yields

$$\left(1 - \partial_y \mathcal{H}_{\tilde{m}_c, w_c}(0,0)\right)\delta C_0(t) = \frac{1}{2}\int_{-\infty}^{+\infty} du\, K_2'(u)\,\delta C_0(t-u).$$
(44)

The above equation does not admit any non-trivial solution, and thus signals a failure in our ansatz. In fact, our analysis rests on the assumption that $\widehat{\delta N}_0(t)$ is of order $O(1)$, which holds only if $\dot{\delta\xi}_0(t)$ is itself of $O(1)$, see for instance Eq. (41) and recall that $h(g) > 0$. Hence, the inconsistency of Eq. (44) shows that $d(\delta\xi(t))/dt$ must vanish close to the transition, or equivalently that the correlation time must diverge at the transition. This also rules out the possibility of a two time-step decay of the correlation function $\delta C_0(t)$, with a fast $O(1)$ timescale and a slow diverging timescale. Indeed, a similar equation would then be obtained for the fast modes only. Note that a diverging timescale in $\delta C_0(t)$ implies that the right-hand side vanishes at the transition, giving

$$1 - \partial_y \mathcal{H}_{\tilde{m}_c, w_c}(0,0) = 0,$$
(45)

which allows us to localize the transition, as we discuss next.

## 5.2  Fixed point phase and the onset of chaos

Equation (45) gives us the onset of chaos. In fact, for $\sigma \le \sigma_c(\mu, \lambda)$, the system reaches a fixed point and $\epsilon = 0$. Thus, for any value of $\sigma$ in this range, one can get the mean and variance of the population sizes in the fixed point phase using Eqs. (33,34). In terms of $\tilde{m}$ and $w$ these equations are written as

$$\frac{1 - \sigma\tilde{m}}{\mu} - \chi_{\tilde{m},w}(0) = \frac{1 - \sigma\tilde{m}}{\mu} - \int_{-\infty}^{+\infty} dg\, P(g)\left(\frac{\sigma g + \sqrt{\sigma^2 g^2 + 4\lambda}}{2}\right) = 0,$$
(46)

and

$$w^2 - \mathcal{H}_{\tilde{m},w}(0,0) = w^2 - \int_{-\infty}^{+\infty} dg\, P(g)\left(\frac{\sigma g + \sqrt{\sigma^2 g^2 + 4\lambda}}{2}\right)^2 = 0,$$
(47)

where $P(g)$ was defined in Eq. (12). At the critical point, Eq. (45) implies that

$$1 - \partial_y \mathcal{H}_{\tilde{m}_c, w_c}(0,0) = 1 - \sigma_c^2 \int_{-\infty}^{+\infty} dg \, P_c(g) \left( \frac{g \sigma_c + \sqrt{4\lambda + g^2 \sigma_c^2}}{2\sqrt{4\lambda + g^2 \sigma_c^2}} \right)^2 = 0, \qquad (48)$$

and provides a third equation. It therefore allows to determine $\tilde{m}_c$, $w_c$ and the position of the critical point $\sigma_c$ when combined with Eqs. (46,47).

## 5.3 Critical regime

We have proven that the correlation time diverges at the transition and we have obtained the position of the critical point. We are now in position to investigate the near-critical regime. We recall the evolution

$$\frac{d\widehat{\delta N}(t)}{dt} = -\left( h(\sigma(g + \epsilon \delta \xi(t))) + \epsilon \widehat{\delta N}(t) \right) \widehat{\delta N}(t) - \sigma f'(\sigma(g + \epsilon \delta \xi(t))) \dot{\delta \xi}(t). \qquad (49)$$

We have seen that $\delta \xi$ must evolve on timescales that diverge close to the transition. If the timescale diverges as $\epsilon^{-r}$, let us define $\tau = \epsilon^r t$. The yet unknown exponent $r > 0$ is such that $\delta \xi'(\tau) = \epsilon^{-r} \dot{\delta \xi}(t = \tau \epsilon^{-r})$ is of order $O(\epsilon^0)$. Later we will prove that $r = 1$, thus entailing the scaling relation $\beta = 2\zeta$ relating the critical exponents introduced in Sec. 2. Upon performing an additional rescaling $\widehat{\delta N}(\tau) \to \epsilon^r \widehat{\delta N}(\tau)$, we get

$$\epsilon^r \widehat{\delta N}'(\tau) = -\left( h(\sigma(g + \epsilon \delta \xi(\tau))) + \epsilon^{r+1} \widehat{\delta N}(\tau) \right) \widehat{\delta N}(\tau) - \sigma f'(\sigma(g + \epsilon \delta \xi(\tau))) \delta \xi'(\tau). \qquad (50)$$

We can now return to the self-consistency equations, which we recall here for clarity. First, Eqs. (33,37), which are the analogues of Eqs. (46,47) in the chaotic phase, allow to derive the static mean and variance of the population sizes $\tilde{m}$ and $w$, at given parameters $\mu, \sigma$ and provided that $\epsilon$ and $\delta C(\tau)$ are known. They read

$$\frac{1 - \sigma \tilde{m}}{\mu} - \chi_{\tilde{m},w}(\epsilon^2) = \epsilon^{r+1} \int_{-\infty}^{+\infty} dg \, P(g) \left\langle \widehat{\delta N}(\tau) \right\rangle_g, \qquad (51)$$

and

$$w^2 = \mathcal{H}_{\tilde{m},w}(\epsilon^2, 0) + \epsilon^2 \mathcal{I}_{\tilde{m},w}[\delta C](\infty), \qquad (52)$$

Then, Eq. (38) provides an equation for the correlation function $\delta C(\tau)$ itself,

$$\delta C(\tau) - \frac{\mathcal{H}_{\tilde{m},w}(\epsilon^2, \epsilon^2 \delta C(\tau)) - \mathcal{H}_{\tilde{m},w}(\epsilon^2, 0)}{\epsilon^2} = \mathcal{I}_{\tilde{m},w}[\delta C](\tau) - \mathcal{I}_{\tilde{m},w}[\delta C](\infty). \qquad (53)$$

The requirement that there exists a solution to Eq. (53) satisfying $\delta C(\tau) = 1$, $\delta C(\infty) = 0$ together with $\delta C'(0) = 0$, meaning that the correlation function is regular at $\tau = 0$, imposes the value of $\epsilon$. We can now evaluate the scaling in $\epsilon$ of the different terms entering the self-consistency equations.

### 5.3.1 Scaling analysis of Eq. (50)

We start by analyzing Eq. (50) and compute the moments of the process $\widehat{\delta N}(\tau)$ entering the self-consistency equations. Because these self-consistency conditions relate the correlations of $\delta \xi(\tau)$ and $\delta N(\tau)$, we get that $\widehat{\delta N}'(\tau)$ is also of order $O(\epsilon^0)$. To leading order, the dynamics Eq. (50) thus reduces to

$$\widehat{\delta N}(\tau) = -\sigma \frac{f'(\sigma g)}{h(\sigma g)} \delta \xi'(\tau) + O(\epsilon) + O(\epsilon^r), \qquad (54)$$

which implies

$$\left\langle \widehat{\delta N}(0)\widehat{\delta N}(\tau) \right\rangle_g = \left( \sigma \frac{f'(\sigma g)}{h(\sigma g)} \right)^2 \left\langle \delta\xi'(\tau)\delta\xi'(0) \right\rangle + O(\epsilon) + O(\epsilon^r). \tag{55}$$

The mean value of $\widehat{\delta N}(\tau)$ in steady-state vanishes to order $O(1)$. In fact, it also vanishes to order $O(\epsilon)$. Indeed, we remark that Eq. (50) yields

$$\left\langle \widehat{\delta N}(\tau)h(g + \epsilon\delta\xi(\tau)) \right\rangle_g + \epsilon^{r+1} \left\langle \widehat{\delta N}(\tau)\widehat{\delta N}(\tau) \right\rangle_g = -\left\langle \sigma f'(\sigma(g + \epsilon\delta\xi(\tau)))\delta\xi'(\tau) \right\rangle.$$

Therefore,

$$h(\sigma g)\left\langle \widehat{\delta N}(\tau) \right\rangle_g + \epsilon\sigma h'(\sigma g)\left\langle \widehat{\delta N}(\tau)\delta\xi(\tau) \right\rangle_g + \epsilon^{r+1}\left\langle \widehat{\delta N}(\tau)\widehat{\delta N}(\tau) \right\rangle_g$$
$$= -\epsilon\,\sigma^2 f''(\sigma g)\left\langle \delta\xi(\tau)\delta\xi'(\tau) \right\rangle + O(\epsilon^2), \tag{56}$$

which, using Eq. (54), gives

$$\left\langle \widehat{\delta N}(\tau) \right\rangle_g = O(\epsilon^2) + O(\epsilon^{r+1}), \tag{57}$$

because $\left\langle \delta\xi(\tau)\delta\xi'(\tau) \right\rangle = \delta C'(0) = 0$ since $\delta C(\tau)$ is an even function. For reasons that will become clear below, we lastly need to evaluate $\left\langle \delta\xi(0)\widehat{\delta N}(\tau) + \delta\xi(\tau)\widehat{\delta N}(0) \right\rangle_g$. Upon rewriting Eq. (50), we get

$$\widehat{\delta N}(\tau) = -\frac{\epsilon^r \widehat{\delta N}'(\tau)}{h(\sigma(g + \epsilon\delta\xi(\tau)))} - \frac{\epsilon^{r+1}}{h(\sigma(g + \epsilon\delta\xi(\tau)))}\widehat{\delta N}(\tau)\widehat{\delta N}(\tau) - \sigma\frac{f'(\sigma(g + \epsilon\delta\xi(\tau)))}{h(\sigma(g + \epsilon\delta\xi(\tau)))}\delta\xi'(\tau). \tag{58}$$

We proceed term by term. First we have

$$\left\langle \delta\xi(0)\frac{\epsilon^r \widehat{\delta N}'(\tau)}{h(\sigma(g + \epsilon\delta\xi(\tau)))} \right\rangle_g = -\epsilon^r\sigma\frac{f'(\sigma g)}{h(\sigma g)^2}\left\langle \delta\xi(0)\delta\xi''(\tau) \right\rangle_g + O(\epsilon^{r+1}) + O(\epsilon^{2r})$$
$$= -\epsilon^r\sigma\frac{f'(\sigma g)}{h(\sigma g)^2}\delta C''(\tau) + O(\epsilon^{r+1}) + O(\epsilon^{2r}). \tag{59}$$

Then

$$\left\langle \delta\xi(0)\frac{\epsilon^{r+1}}{h(\sigma(g + \epsilon\delta\xi(\tau)))}\widehat{\delta N}(\tau)\widehat{\delta N}(\tau) \right\rangle_g = O(\epsilon^{r+2}),$$

as the leading order is the product of an odd number of zero-mean Gaussian variables. Finally, the last term in Eq. (58) could appear to dominate the correlation $\left\langle \delta\xi(0)\widehat{\delta N}(\tau) + \delta\xi(\tau)\widehat{\delta N}(0) \right\rangle_g$ but it doesn't due to the $\tau \to -\tau$ symmetry. Indeed

$$\left\langle \delta\xi(0)\frac{f'(\sigma(g + \epsilon\delta\xi(\tau)))}{h(\sigma(g + \epsilon\delta\xi(\tau)))}\delta\xi'(\tau) + \delta\xi(\tau)\frac{f'(\sigma(g + \epsilon\delta\xi(0)))}{h(\sigma(g + \epsilon\delta\xi(0)))}\delta\xi'(0) \right\rangle_g$$
$$= \frac{f'(\sigma g)}{h(\sigma g)}\left\langle \delta\xi(0)\delta\xi'(\tau) + \delta\xi(\tau)\delta\xi'(0) \right\rangle + O(\epsilon^2) = O(\epsilon^2). \tag{60}$$

Therefore,

$$\left\langle \delta\xi(0)\widehat{\delta N}(\tau) + \delta\xi(\tau)\widehat{\delta N}(0) \right\rangle_g = 2\epsilon^r\sigma_c\frac{f'(\sigma_c g)}{h(\sigma_c g)^2}\delta C''(\tau) + O(\epsilon^2) + O(\epsilon^{2r}), \tag{61}$$

where to leading order we have replaced $\sigma$ by its value at the critical point.

### 5.3.2 Scaling analysis of Eq. (53)

We can now analyze the small $\epsilon$ behavior of Eq. (53) which requires computing $\mathcal{I}_{\tilde{m},w}[\delta C](\tau) - \mathcal{I}_{\tilde{m},w}[\delta C](\infty)$. By definition, it reads

$$
\mathcal{I}_{\tilde{m},w}[\delta C](\tau) - \mathcal{I}_{\tilde{m},w}[\delta C](\infty) = \epsilon^{-1+r} \int_{-\infty}^{+\infty} \mathrm{d}g \, P(g) \Big\langle f\left(\sigma(g + \epsilon \delta \xi(0))\right) \widehat{\delta N}(\tau) + f\left(\sigma(g + \epsilon \delta \xi(\tau))\right) \widehat{\delta N}(0) \Big\rangle_g
$$
$$
- 2\epsilon^{-1+r} \int_{-\infty}^{+\infty} \mathrm{d}g \, P(g) \left\langle f\left(\sigma(g + \epsilon \delta \xi)\right)\right\rangle \left\langle \widehat{\delta N} \right\rangle_g + \epsilon^{2r} \int_{-\infty}^{+\infty} \mathrm{d}g \, P(g) \left[ \left\langle \widehat{\delta N}(0)\widehat{\delta N}(\tau) \right\rangle_g - \left\langle \widehat{\delta N} \right\rangle_g^2 \right].
$$

Using Eqs. (55, 57) we first get,

$$
\epsilon^{2r} \int_{-\infty}^{+\infty} \mathrm{d}g \, P(g) \left[ \left\langle \widehat{\delta N}(0)\widehat{\delta N}(\tau) \right\rangle_g - \left\langle \widehat{\delta N} \right\rangle_g^2 \right]
$$
$$
= \epsilon^{2r} \left\langle \delta \xi'(\tau) \delta \xi'(0) \right\rangle \int_{-\infty}^{+\infty} \mathrm{d}g \, P(g) \left( \sigma \frac{f'(\sigma g)}{h(\sigma g)} \right)^2 + O\left(\epsilon^{3r}\right) + O\left(\epsilon^{1+2r}\right). \tag{62}
$$

Furthermore, we obtain

$$
\Big\langle f\left(\sigma(g + \epsilon \delta \xi(0))\right) \widehat{\delta N}(\tau) + f\left(\sigma(g + \epsilon \delta \xi(\tau))\right) \widehat{\delta N}(0) \Big\rangle - 2 \left\langle f\left(\sigma(g + \epsilon \delta \xi)\right)\right\rangle_g \left\langle \widehat{\delta N} \right\rangle_g
$$
$$
= \frac{\epsilon^2}{2} \sigma^2 f''(\sigma g) \Big\langle (\delta \xi(0)^2 - 1)\widehat{\delta N}(\tau) + (\delta \xi(\tau)^2 - 1)\widehat{\delta N}(0) \Big\rangle_g
$$
$$
+ \epsilon \sigma f'(\sigma g) \Big\langle \delta \xi(0)\widehat{\delta N}(\tau) + \delta \xi(\tau)\widehat{\delta N}(0) \Big\rangle_g + O\left(\epsilon^3\right). \tag{63}
$$

From Eq. (61), it appears that the first term in the right-hand side is of order $O(\epsilon^{r+1})$. The second term is subleading since

$$
\Big\langle (\delta \xi(0)^2 - 1)\widehat{\delta N}(\tau) + (\delta \xi(\tau)^2 - 1)\widehat{\delta N}(0) \Big\rangle_g = O(\epsilon).
$$

In fact, to leading order, this expression only involves products of an odd number of zero-mean Gaussian variables. Therefore,

$$
\mathcal{I}_{\tilde{m},w}[\delta C](\tau) - \mathcal{I}_{\tilde{m},w}[\delta C](\infty) = \epsilon^{2r} \delta C''(\tau) \int_{-\infty}^{+\infty} \mathrm{d}g \, P(g) \left( \sigma \frac{f'(\sigma g)}{h(\sigma g)} \right)^2 + O\left(\epsilon^{3r}\right) + O\left(\epsilon^{2+r}\right),
$$

since $\left\langle \delta \xi'(\tau) \delta \xi'(0) \right\rangle = -\delta C''(\tau)$. Finally, we expand the left-hand side of Eq. (53) to obtain

$$
\delta C(\tau) - \frac{\mathcal{H}_{\tilde{m},w}\left(\epsilon^2, \epsilon^2 \delta C(\tau)\right) - \mathcal{H}_{\tilde{m},w}\left(\epsilon^2, 0\right)}{\epsilon^2} = \delta C(\tau)\left(1 - \partial_y \mathcal{H}_{\tilde{m},w}\left(\epsilon^2, 0\right)\right)
$$
$$
- \frac{1}{2}\epsilon^2 \delta C(\tau)^2 \partial_y^2 \mathcal{H}_{\tilde{m},w}(0,0) + O\left(\epsilon^4\right). \tag{64}
$$

We now use the equation for the onset of chaos Eq. (48) to get

$$
\delta C(\tau) - \frac{\mathcal{H}_{\tilde{m},w}\left(\epsilon^2, \epsilon^2 \delta C(\tau)\right) - \mathcal{H}_{\tilde{m},w}\left(\epsilon^2, 0\right)}{\epsilon^2} = \delta C(\tau)\left(\partial_y \mathcal{H}_{\tilde{m}_c,w_c}(0,0) - \partial_y \mathcal{H}_{\tilde{m},w}\left(\epsilon^2, 0\right)\right)
$$
$$
- \frac{1}{2}\epsilon^2 \delta C(\tau)^2 \partial_y^2 \mathcal{H}_{\tilde{m}_c,w_c}(0,0) + O\left(\epsilon^4\right),
$$
$$
= \delta C(\tau)\left(\partial_y \mathcal{H}_{\tilde{m}_c,w_c}(0,0) - \partial_y \mathcal{H}_{\tilde{m},w}(0,0)\right) - \delta C(\tau)\epsilon^2 \partial_x \partial_y \mathcal{H}_{\tilde{m}_c,w_c}(0,0)
$$
$$
+ \frac{1}{2}\epsilon^2 \delta C(\tau)^2 \partial_y^2 \mathcal{H}_{\tilde{m}_c,w_c}(0,0) + O\left(\epsilon^4\right). \tag{65}
$$

Therefore we obtain the equation satisfied by the correlation function in the vicinity of the critical point,

$$-\delta C(\tau)\epsilon^2 \partial_x \partial_y \mathcal{H}_{\tilde{m}_c,w_c}(0,0) + \delta C(\tau)\left(\partial_y \mathcal{H}_{\tilde{m}_c,w_c}(0,0) - \partial_y \mathcal{H}_{\tilde{m},w}(0,0)\right) - \frac{1}{2}\epsilon^2 \delta C(\tau)^2 \partial_y^2 \mathcal{H}_{\tilde{m}_c,w_c}(0,0) + O\left(\epsilon^4\right)$$

$$= \epsilon^{2r}\delta C''(\tau)\int_{-\infty}^{+\infty} \mathrm{d}g\, P(g)\left(\sigma\frac{f'(\sigma g)}{h(\sigma g)}\right)^2 + O\left(\epsilon^{3r}\right) + O\left(\epsilon^{2+r}\right).$$

In order for a physically sound solution to exist, all leading order terms must have the same scaling as $\epsilon \to 0$. This imposes the scaling behavior $r = 1$. In the following, we use the notations

$$\gamma = -\partial_x \partial_y \mathcal{H}_{\tilde{m}_c,w_c}(0,0) + \lim_{\sigma \to \sigma_c} \frac{\partial_y \mathcal{H}_{\tilde{m}_c,w_c}(0,0) - \partial_y \mathcal{H}_{\tilde{m},w}(0,0)}{\epsilon^2},$$

$$\kappa = \frac{1}{2}\partial_y^2 \mathcal{H}_{\tilde{m}_c,w_c}(0,0), \tag{66}$$

$$\omega = \int_{-\infty}^{+\infty} \mathrm{d}g\, P_c(g)\left(\sigma_c\frac{f'(\sigma_c g)}{h(\sigma_c g)}\right)^2.$$

$\kappa$ and $\omega$ are determined by the known value of $w_c$ and $\tilde{m}_c$ at the transition. The parameter $\gamma$ remains unknown as finding it requires to investigate the behavior of $w$ and $\tilde{m}$ close to the transition, as will be done in the next section. To leading order, the correlation function obeys

$$\delta C_0''(\tau) = \frac{\gamma}{\omega}\delta C_0(\tau) - \frac{\kappa}{\omega}\delta C_0(\tau)^2.$$

The above equation can be seen as the classical equation of motion of a massive particle in a cubic potential. The conditions $\delta C_0(0) = 1$, $\delta C_0'(0) = 0$ and $\delta C_0(\infty) = 0$ then constrain the admissible value of $\gamma$ to be

$$\gamma = \frac{2\kappa}{3},$$

with $\kappa > 0$. In terms of the known constants $\kappa$ and $\omega$, the solution reads

$$\delta C_0(\tau) = 1 - \mathrm{Tanh}^2\left(\frac{\sqrt{\kappa}\tau}{6\sqrt{\omega}}\right). \tag{67}$$

The amplitude of the fluctuations $\epsilon$, and therefore the critical exponent $\beta$, are derived from the constraint on $\gamma$. As mentioned above, this requires to investigate the behavior of $\tilde{m}$ and $w$ in the vicinity of the critical point, as we do in Sec. 5.3.3. Equation (67) completes the derivation of the scaling form in Eq. (7). We give the expression of the coefficient $\tau_c(\lambda,\mu)$ entering Eq. (7) at the end of Sec. 5.3.3.

### 5.3.3 Scaling analysis of Eq. (51, 52)

We recall that $\left\langle\widehat{\delta N}(\tau)\right\rangle_g = O\left(\epsilon^2\right)$, see Eq. (57). Neglecting all corrections beyond order $\epsilon^2$, Eq. (51) becomes

$$\frac{1-\sigma\tilde{m}}{\mu} - \chi_{\tilde{m},w}(0) - \epsilon^2\chi_{\tilde{m}_c,w_c}'(0) = \frac{1-\sigma\tilde{m}}{\mu} - \int_{-\infty}^{+\infty}\mathrm{d}g\, P(g)f(\sigma g) - \frac{1}{2}\epsilon^2\sigma_c^2\int_{-\infty}^{+\infty}\mathrm{d}g\, P_c(g)f''(\sigma_c g) = 0.$$
$$\tag{68}$$

Additionally, we note that

$$\mathcal{I}_{\tilde{m},w}[\delta C](\infty) = 2\int_{-\infty}^{+\infty}\mathrm{d}g\, P(g)\left\langle\bar{N}\right\rangle_g\left\langle\widehat{\delta N}\right\rangle_g + \epsilon^2\int_{-\infty}^{+\infty}\mathrm{d}g\, P(g)\left\langle\widehat{\delta N}\right\rangle_g^2 = O\left(\epsilon^2\right).$$

Therefore, neglecting all corrections beyond order $\epsilon^2$, Eq. (52) becomes

$$w^2 - \int_{-\infty}^{+\infty} \mathrm{d}g \, P(g) f(\sigma g)^2 - \epsilon^2 \sigma_c^2 \int_{-\infty}^{+\infty} \mathrm{d}g \, P_c(g) f(\sigma_c g) f''(\sigma_c g) = 0. \tag{69}$$

Lastly, we recall the constraint $\gamma = 2\kappa/3$ which imposes

$$\lim_{\sigma \to \sigma_c} \frac{\partial_y \mathcal{H}_{\tilde{m}_c, w_c}(0,0) - \partial_y \mathcal{H}_{\tilde{m}, w}(0,0)}{\epsilon^2} = \frac{1}{3} \partial_y^2 \mathcal{H}_{\tilde{m}_c, w_c}(0,0) + \partial_x \partial_y \mathcal{H}_{\tilde{m}_c, w_c}(0,0). \tag{70}$$

We can now expand Eqs. (70,68,69) in terms of the small parameter $x$ defined as the distance to the critical point, $x \equiv \sigma - \sigma_c$. To linear level, $w = w_c + x w_1$ and $\tilde{m} = \tilde{m}_c + x \tilde{m}_1$, from which it is clear that Eq. (70) requires the scaling form $\epsilon = \epsilon_0 \sqrt{x}$. This entails the value of the critical exponent $\beta = 1$, from which it follows that $\zeta = 1/2$. The expansion of Eqs. (70,68,69) then fixes the values of $w_1$, $\tilde{m}_1$ and $\epsilon_0$. We get from Eqs. (68,69)

$$-\tilde{m}_1 \left( \frac{\sigma_c}{\mu} + \frac{1}{w_c^2} \int_{-\infty}^{+\infty} \mathrm{d}g \, P_c(g)(g - \tilde{m}_c) f(\sigma_c g) \right) - \frac{w_1}{w_c^3} \int_{-\infty}^{+\infty} \mathrm{d}g \, P_c(g) \left[ (g - \tilde{m}_c)^2 - w_c^2 \right] f(\sigma_c g)$$

$$= \frac{\tilde{m}_c}{\mu} + \int_{-\infty}^{+\infty} \mathrm{d}g \, P_c(g) \, g f'(\sigma_c g) + \frac{1}{2} \epsilon_0^2 \sigma_c^2 \int_{-\infty}^{+\infty} \mathrm{d}g \, P_c(g) f''(\sigma_c g), \tag{71}$$

and

$$w_1 \left( 2 w_c - \frac{1}{w_c^3} \int_{-\infty}^{+\infty} \mathrm{d}g \, P_c(g) \left[ (g - \tilde{m}_c)^2 - w_c^2 \right] f(\sigma_c g)^2 \right) - \frac{\tilde{m}_1}{w_c^2} \int_{-\infty}^{+\infty} \mathrm{d}g \, P_c(g)(g - \tilde{m}_c) f(\sigma_c g)^2$$

$$= \int_{-\infty}^{+\infty} \mathrm{d}g \, P_c(g) 2 g f(\sigma_c g) f'(\sigma_c g) + \epsilon_0^2 \sigma_c^2 \int_{-\infty}^{+\infty} \mathrm{d}g \, P_c(g) f(\sigma_c g) f''(\sigma_c g). \tag{72}$$

Lastly, from Eq. (70), we obtain,

$$\left( \sigma_c^4 \int_{-\infty}^{+\infty} \mathrm{d}g \, P_c(g) f^{(3)}(g) f'(g) + \frac{1}{3} \sigma_c^4 \int_{-\infty}^{+\infty} \mathrm{d}g \, P_c(g) f''(g)^2/3 \right) \epsilon_0^2 = -\frac{2\sigma_c \int_{-\infty}^{+\infty} \mathrm{d}g \, P_c(g) f'(g\sigma_c)^2}{}$$

$$+ \sigma_c^2 \int_{-\infty}^{+\infty} \mathrm{d}g \, P_c(g) \left( \tilde{m}_1 \frac{(g - \tilde{m}_c)}{w_c^2} + w_1 \frac{((g - \tilde{m}_c)^2 - w_c^2)}{w_c^3} \right) f'(g\sigma_c)^2$$

$$+ \sigma_c^2 \int_{-\infty}^{+\infty} \mathrm{d}g \, P_c(g) 2 g f''(\sigma_c g) f'(\sigma_c g). \tag{73}$$

The parameter $\epsilon_0$ inferred from these equations is directly related to the coefficient $Q_c(\lambda, \mu)$ of Eq. (3) through $Q_c(\lambda, \mu) = \epsilon_0^2$. It is also related to the coefficient $\tau_c(\lambda, \mu)$ of Eq. (7) which can be read from Eq. (67)

$$\tau_c(\lambda, \mu) = \frac{6\sqrt{\omega}}{\epsilon_0 \sqrt{\kappa}}, \tag{74}$$

after recalling that $\tau = \epsilon t$.

## 5.4 Suppression of the chaotic phase at large $\lambda$

These results indicate that the chaotic phase is suppressed for large values of $\lambda$, above a critical value $\lambda_c(\mu)$. For $\lambda > \lambda_c(\mu)$, we find that the collective dynamics in Eq. (1) directly experiences

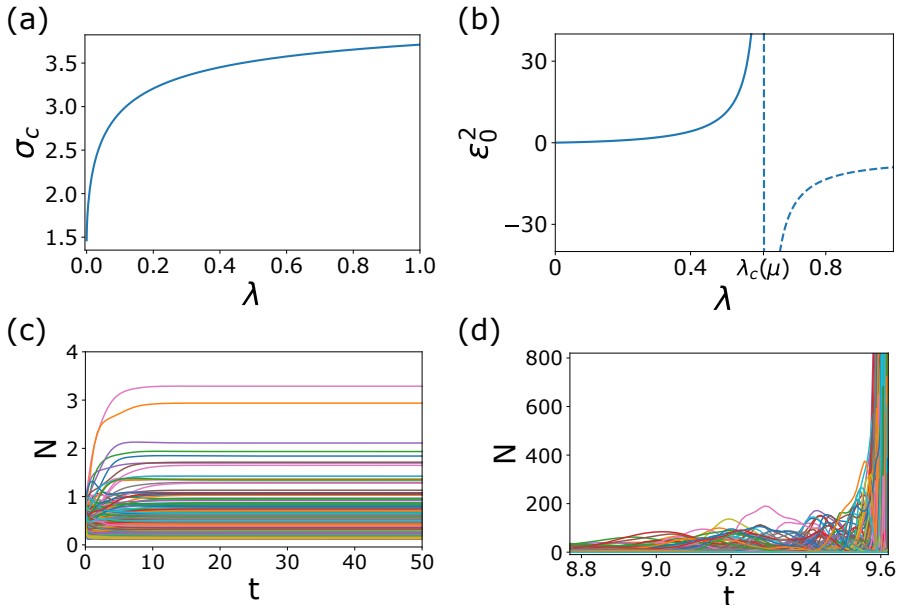

Figure 7: **Suppression of the chaotic phase at large migration rate. (a)** Onset of stability of the fixed point phase $\sigma_c(\lambda, \mu)$ as a function of $\lambda$ for $\mu = 10$. **(b)** Amplitude of the chaotic fluctuations close to the transition $Q_c(\lambda, \mu) = \epsilon_0^2$ as a function of $\lambda$ for $\mu = 10$ as predicted by the linear system of Eqs. (71,72,73). Above a certain value $\lambda > \lambda_c(\mu) \simeq 0.6$, the predicted value for $Q_c(\lambda, \mu)$ is negative, therefore signaling the suppression of the chaotic phase. **(c, d)** Many-body simulations of Eq. (1) with $S = 4000$ for $\mu = 10$ and $\lambda = 0.8 > \lambda_c(\mu)$. The loss of stability of the fixed phase is predicted to take place around $\sigma_c(\lambda, \mu) \simeq 0.65$. **(c)** Simulations at $\sigma = 0.6 < \sigma_c(\lambda, \mu)$. The dynamics converges to a fixed point. **(d)** Simulations at $\sigma = 0.7 > \sigma_c(\lambda, \mu)$. The dynamics enters a regime of unbounded growth of the population sizes.

a transition from the fixed point phase when $\sigma < \sigma_c(\lambda, \mu)$ to a phase of unbounded growth when $\sigma > \sigma_c(\lambda, \mu)$, see Fig. 7 below. This phase of unbounded growth, which corresponds to blow-up of population sizes, was already investigated when the migration is low and was found to appear for large values of $\sigma$ [3].

The critical parameter $\sigma_c(\lambda, \mu)$ which describes the onset of stability loss of the fixed point solution is still given by the solution of Eqs. (46,47,48). However, the set of linear equations prescribing the steady-state amplitude of the fluctuations close to the critical point $\epsilon_0^2 = Q_c(\lambda, \mu)$, see Eqs. (71,72,73), admits a diverging solution at $\lambda = \lambda_c(\mu)$ and only (unphysical) negative solutions for $\lambda > \lambda_c(\mu)$.

## 5.5   Results when $\lambda \ll 1$

We have characterized the near-critical regime when $\lambda > 0$ is finite. The values of $\tilde{m}_c$, $w_c$, and $\sigma_c$ can be determined numerically for any finite $\lambda$ and from there the values of $w_1, \tilde{m}_1$ and $\epsilon_0$ can also be determined. In the following, we focus in the regime where $\lambda \ll 1$ which has been the regime of interest of most of the numerical and analytical literature so far [20, 21, 24, 25], and for which we make further analytical progress. Our results demonstrate the existence of the crossover discussed in Sec. 2.3, as shown by determining $Q_c(\lambda, \mu)$ and $\tau_c(\lambda, \mu)$ when $\lambda \ll 1$.

### 5.5.1 Critical line

We start with the equations yielding $\tilde{m}_c$, $w_c$, and $\sigma_c$. They read

$$1 - \frac{\sigma_c^2}{\sqrt{2\pi w_c^2}} \int_{-\infty}^{+\infty} \mathrm{d}\xi \exp\left(-\frac{\xi^2}{2w_c^2}\right) f'(\sigma_c(\xi + \tilde{m}_c))^2 = 0, \tag{75}$$

$$\frac{1 - \sigma \tilde{m}_c}{\mu} - \frac{1}{\sqrt{2\pi w_c^2}} \int_{-\infty}^{+\infty} \mathrm{d}\xi \exp\left(-\frac{\xi^2}{2w_c^2}\right) f(\sigma_c(\xi + \tilde{m}_c)) = 0, \tag{76}$$

$$w_c^2 - \frac{1}{\sqrt{2\pi w_c^2}} \int_{-\infty}^{+\infty} \mathrm{d}\xi \exp\left(-\frac{\xi^2}{2w_c^2}\right) f(\sigma_c(\xi + \tilde{m}_c))^2 = 0. \tag{77}$$

with

$$f(x) = \frac{x + \sqrt{x^2 + 4\lambda}}{2}.$$

We recall that $\lim_{\lambda\to 0} \tilde{m}_c = 0$ and so we rewrite $\tilde{m}_c \to \sqrt{\lambda}\delta m_c$. We also recall $\lim_{\lambda\to 0} \sigma_c = \sqrt{2}$ and $\lim_{\lambda\to 0} w_c = \sqrt{\pi}/\mu$. This allows us to expand Eq. (75) as

$$0 = 1 - \frac{\sigma_c^2}{\sqrt{2\pi w_c^2}} \int_{-\infty}^{+\infty} \mathrm{d}\xi \exp\left(-\frac{\xi^2}{2w_c^2}\right) f'\left(\sigma_c\left(\xi + \sqrt{\lambda}\delta m_c\right)\right)^2,$$

$$= 1 - \frac{\sigma_c^2}{2} - \frac{\sigma_c^2 \sqrt{\lambda}}{\sqrt{2\pi w_c^2}} \int_{-\infty}^{+\infty} \mathrm{d}\xi \exp\left(-\frac{\lambda\xi^2}{2w_c^2}\right) \left[\frac{1}{4}\left(1 + \frac{\sigma_c(\xi + \delta m_c)}{\sqrt{\sigma_c^2(\xi + \delta m_c)^2 + 4}}\right)^2 - \Theta(\xi)\right],$$

$$= 1 - \frac{\sigma_c^2}{2} - \sqrt{\lambda}\frac{\sqrt{2}\mu}{\pi} \int_{-\infty}^{+\infty} \mathrm{d}\xi \left[\frac{1}{4}\left(1 + \frac{(\xi + \delta m_c)}{\sqrt{(\xi + \delta m_c)^2 + 2}}\right)^2 - \Theta(\xi)\right] + o\left(\sqrt{\lambda}\right),$$

$$= 1 - \frac{\sigma_c^2}{2} - \sqrt{\lambda}\frac{\sqrt{2}\mu}{\pi}\left(\delta m_c - \frac{\pi}{2\sqrt{2}}\right) + o\left(\sqrt{\lambda}\right).$$

We therefore expand $\sigma_c = \sqrt{2} + \sqrt{\lambda}\delta\sigma_c$ and $w_c = \sqrt{\pi}/\mu + \sqrt{\lambda}\delta w_c$ to obtain to leading order

$$\delta\sigma_c = \frac{\mu}{\pi}\left(\frac{\pi}{2\sqrt{2}} - \delta m_c\right). \tag{78}$$

We proceed similarly to expand Eq. (77)

$$0 = w_c^2 - \frac{1}{\sqrt{2\pi w_c^2}} \int_{-\infty}^{+\infty} \mathrm{d}\xi \exp\left(-\frac{\xi^2}{2w_c^2}\right) f(\sigma_c(\xi + \tilde{m}_c))^2,$$

$$= w_c^2 - \frac{1}{\sqrt{2\pi w_c^2}} \int_{-\infty}^{+\infty} \mathrm{d}\xi \exp\left(-\frac{\xi^2}{2w_c^2}\right) \left[\sigma_c^2\left(\xi + \sqrt{\lambda}\delta m_c\right)^2 + 2\lambda\right]$$

$$- \frac{\lambda^{3/2}}{\sqrt{2\pi w_c^2}} \int_{-\infty}^{+\infty} \mathrm{d}\xi \exp\left(-\frac{\lambda\xi^2}{2w_c^2}\right) \left[\frac{1}{4}\left(\sigma_c(\xi + \delta m_c) + \sqrt{\sigma_c^2(\xi + \delta m_c)^2 + 4}\right)^2 - \left[\sigma_c^2(\xi + \delta m_c)^2 + 2\right]\Theta(\xi)\right],$$

$$= w_c^2 - \frac{1}{\sqrt{2\pi w_c^2}} \int_0^{+\infty} \mathrm{d}\xi \exp\left(-\frac{\xi^2}{2w_c^2}\right) \left[\sigma_c^2\left(\xi + \sqrt{\lambda}\delta m_c\right)^2 + 2\lambda\right] + O\left(\lambda^{3/2}\right),$$

$$= w_c^2\left(1 - \frac{\sigma_c^2}{2}\right) - \left(\sqrt{\lambda}\sqrt{\frac{2}{\pi}}\delta m_c w_c \sigma_c^2 + \lambda + \frac{\lambda}{2}\sigma_c^2\delta m_c^2\right) + O\left(\lambda^{3/2}\right).$$

Therefore, to leading order, we obtain

$$\delta m_c = -\frac{\pi}{2\pi}\delta\sigma_c. \tag{79}$$

Hence, Eqs. (78,79) allow to find the shift in the location of the critical point as

$$\delta\sigma_c \simeq \frac{\mu}{\sqrt{2}},$$
$$\delta m_c \simeq -\frac{\pi}{2\sqrt{2}}.$$

Lastly, $\delta w_c$ can be obtained to leading order by expanding Eq. (76)

$$
\begin{aligned}
0 &= \frac{1-\sqrt{\lambda}\sigma_c\delta m_c}{\mu} - \frac{1}{\sqrt{2\pi w_c^2}}\int_{-\infty}^{+\infty}\mathrm{d}\xi\exp\left(-\frac{\xi^2}{2w_c^2}\right)f\left(\sigma_c\left(\xi+\sqrt{\lambda}\delta m_c\right)\right)\\
&= \frac{1-\sqrt{\lambda}\sigma_c\delta m_c}{\mu} - \frac{\sigma_c}{\sqrt{2\pi w_c^2}}\int_0^{+\infty}\mathrm{d}\xi\exp\left(-\frac{\xi^2}{2w_c^2}\right)\left(\xi+\sqrt{\lambda}\delta m_c\right)\\
&\quad -\frac{\lambda}{\sqrt{2\pi w_c^2}}\int_{-\infty}^{+\infty}\mathrm{d}\xi\exp\left(-\frac{\lambda\xi^2}{2w_c^2}\right)\left[\frac{\sqrt{4+\sigma_c^2(\xi+\delta m_c)^2}}{2}-\sigma_c\xi\,\Theta(\xi)\right].
\end{aligned}
$$

We note that the last term scales as $O(\lambda\ln\lambda)$ since

$$\frac{\sqrt{4+\sigma_c^2(\xi+\delta m_c)^2}}{2}-\sigma_c\xi\,\Theta(\xi) =_{x\to\pm\infty} O\left(\frac{1}{x}\right).$$

Therefore, we obtain to leading order

$$0 = \frac{1-\sqrt{\lambda}\sigma_c\delta m_c}{\mu} - \sigma_c\left(\frac{w_c}{\sqrt{2\pi}}+\frac{\sqrt{\lambda}\delta m_c}{2}\right) + o\left(\sqrt{\lambda}\right),$$

which yields

$$\delta w_c \simeq \frac{\sqrt{\pi}}{4\mu}\left(\pi(\mu+2)-2\mu\right).$$

### 5.5.2 Near-critical regime

We now turn to the determination of $Q_c(\lambda,\mu)$ and $\tau_c(\lambda,\mu)$. We recall that the leading order corrections close to the critical point $\tilde{m}_1$, $w_1$ and $\epsilon_0$ satisfy the set of linear equations given in Eqs. (71,72,73). After some lengthy but straightforward algebra, we obtain the leading order expression of these coefficients. Crucially, the amplitude of the fluctuations vanishes when $\lambda\to 0$ as we find

$$Q_c(\lambda,\mu) = \epsilon_0^2 \simeq \frac{16\sqrt{\lambda}}{\sqrt{2}\mu}. \tag{80}$$

This suggests that the near-critical regime is controlled by another scaling limit if the migration rate $\lambda$ goes to 0 faster than $\sigma$ goes to the critical value $\sigma_c(\lambda=0,\mu)=\sqrt{2}$, as discussed in Sec. 2.3. It is also instructive to look at the behavior of timescales when $\lambda\ll 1$. The expression of the timescale $\tau_c(\lambda,\mu)$ entering Eq. (67) was given in Eq. (74). We find that to leading order as $\lambda\to 0^+$, $\omega/\kappa\simeq 4/3$, so that we get

$$\tau_c(\lambda,\mu) \simeq \frac{\sqrt{3\mu\sqrt{2}}}{\lambda^{1/4}}, \tag{81}$$

which diverges as $\lambda\to 0$.

# 6 Conclusion

We have analytically described the Lotka-Volterra dynamics with many species and random interactions between them in the vicinity of the critical point separating the fixed phase to a phase of perpetual fluctuations. When approaching the critical point from the fluctuating phase, timescales are large and diverge at the transition (critical slowing down), while the size of the temporal fluctuations decreases continuously to zero. To characterize these two effects, we obtain the scaling behavior of the correlation function near the critical point. We identify two critical exponents $\beta$ and $\zeta$ in the scaling theory, and calculate their values.

Our study highlights the effect of the migration rate $\lambda$ on the critical dynamics. Depending on $\lambda$ and the distance from the critical point $\sigma - \sigma_c$, we identify three different scaling regimes: one for $\lambda = 0$, one for $\lambda > 0$ fixed, and one for $\lambda \to 0^+$, meaning that $\sqrt{\lambda} \ll \sigma - \sigma_c \ll 1$. This third regime $\sqrt{\lambda} \ll \sigma - \sigma_c$ is commonly probed in numerical investigations of the dynamics [20, 24]. The scaling behavior and the values of the exponents are different between these regimes.

This work raises a number of interesting questions for future study. One is the study of critical behavior when approaching the transition from the fixed point phase. In this phase there are no endogenous dynamics at long times, but one can consider the relaxation close to the fixed point, and the response to external noise. Previous works have considered the linearized dynamics around the fixed point [2, 20] by only looking at the surviving species (those with $N > 0$ at the fixed point reached when $\lambda \to 0^+$). Yet the present and previous works [21, 22] highlight the importance of "species turnover" events where species are exchanged between $O(\lambda)$ and $O(1)$ values, and this non-linear effect might be relevant also on the fixed point side of the transition. Another interesting direction is the behavior at finite number of species $S$. The question of the width of the crossover region between the fixed-point phase and the chaotic or aging ones in finite size systems remains open, as is the behavior in this region; Simulations show that close to the transition limit cycles are sometimes reached, even with hundreds of variables.

Finally, it would be interesting to see if any of the critical behavior could be observed in experiments [19] or field studies. The main qualitative phenomena–large timescales and small temporal fluctuations near the transition–are promising candidates.

# Acknowledgements

We thank Anna Frishman and Yanay Tovi for useful discussions.

# A Numerical methods

Here we detail the numerical procedures used to solve the DMFT equations. We start by giving detail about the $\lambda > 0$ case. We recall the DMFT equations

$$\dot{N}(t) = N(t)[1 - N(t) - \mu m(t) + \sigma \xi(t)] + \lambda, \tag{82}$$

with the conditions

$$m(t) = \langle N(t) \rangle, \tag{83}$$

and

$$C(t, t') = \langle \xi(t)\xi(t') \rangle = \langle N(t)N(t') \rangle, \tag{84}$$

These are self-consistent equations. The trajectory $N(t)$ depends on $\xi(t)$, which is sampled from the correlation function $C(t, t')$, and the function $m(t)$. Self-consistently, $C(t, t)$, $m(t)$ depend on the statistics of $N(t)$, see Eqs. (83, 84). This self-consistency is standard in DMFT formulations. We used a well-known numerical method to solve it [20, 26]. It starts with a guess for $C(t, t')$, $m(t)$, generates realizations of $\xi(t)$, and from that trajectories $N(t)$, which are then used to update $C(t, t')$, $m(t)$. This is repeated until convergence.

In practice, at small $\delta\sigma = \sigma - \sigma_c$, the DMFT simulations were implemented with some theoretical knowledge the expected outcome. Let $\tau_{\exp}$ be an expectation for the correlation time in steady-state at finite $\delta\sigma$. Here we used $\tau_{\exp} = \tau_c(\lambda, \mu)/\sqrt{\delta\sigma}$. For each iteration, the noise $\xi(t)$ was sampled from the correlation function $C(t, t')$ over a time interval $t \in [0, 100\tau_{\exp}]$. The time interval was discretized in such a way that the bining $dt$ becomes smaller and smaller with time. Here we used $dt = 0.048\tau_{\exp}$ for $t \in [0, 60\tau_{\exp}]$, $dt = 0.0075\tau_{\exp}$ for $t \in [60\tau_{\exp}, 90\tau_{\exp}]$ and $dt = 0.0033\tau_{\exp}$ for $t \in [90\tau_{\exp}, 100\tau_{\exp}]$. For each realization of the dynamics in Eq. (9), the population size was initialized at a near fixed point value, so that $(N(0) - 0.01)[1 - (N(0) - 0.01) - \mu m(0) + \sigma\xi(0)] + \lambda = 0$. For the first iteration, leveraging on our estimates for fluctuations and timescales, the initial guess for the correlation function $C(t, t')$ and mean $m(t)$ were the following: $C(t, t') = w_c^2 + \delta\sigma\exp(-|t - t'|/\tau_{\exp})$ and $m(t) = m_c$, which are small perturbations around their values at the critical point. We then used (i) 150 iterations with averaging over 1 000 realizations and injection fraction 0.3 followed by (ii) 40 iterations with averaging over 10 000 realizations and injection fraction 0.3 followed by (iii) 400 iterations with averaging over 10 000 realizations and injection fraction 0.03 followed by (iv) 300 iterations with averaging over 10 000 realizations and injection fraction 0.003 followed by (v) 100 iterations with averaging over 100 000 realizations and injection fraction 0.003 and followed by (vi) 10 iterations with averaging over 1 000 000 realizations and injection fraction 0.003.

The simulations in rescaled time for the cases $\lambda \to 0^+$ and $\lambda = 0$, corresponding to the self-consistency Eqs. (13,14,15,16) and Eqs. (14,15,16,29) respectively, used a very similar protocol (upon replacing $t$ by $s$). For $\lambda > 0$, we chose $\tau_{\exp} = 2$ and $\tau_{\exp} = 2/\delta\sigma$ for $\lambda \to 0^+$. For the first iteration, we also chose $C(s, s') = w_c^2 + \delta\sigma^2\exp(-|s - s'|/\tau_{\exp})$. Lastly, the initial condition for the variable $z(s)$ at the beginning of each realization of the dynamics was $z(s) = -1$ if $1 - \mu m(0) + \sigma\xi(0) < 0$ and $z(s) = 0$ otherwise.

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
