# Peer review of "Critical behavior of a phase transition in the dynamics of interacting populations"

_SciPost Physics_

## Round 2 · Referee Report · Anonymous (Referee 1) · 2024-5-27

Strengths

  1. Thorough analytical aramework. The authors employ the Dynamical Mean-Field Theory to establish a robust theoretical framework.

  2. Wide-ranging research interest with the identification of different Universality Classes. It enhances the understanding of how migration impacts the critical behaviour of interacting populations and, more broadly, complex heterogeneous systems.

Weaknesses

  1. Technicality. The paper comes across as too technical in several sections and for an extremely specialised community.

  2. Extreme concision of the numerical part. This section would benefit from more details to validate the robustness and broader applicability of the results.

Report

The paper "Critical behavior of a phase transition in the dynamics of interacting populations" explores the intricate dynamics of many-variable differential equations with random interactions, focusing on ecological systems modeled by Lotka-Volterra equations. This study provides a comprehensive analytical description and identifies three regimes depending on migration rate strength, along with the associated critical exponents in the scaling theory. The analytical rigor of the paper is commendable, substantiated by detailed derivations. Furthermore, the identification of three different universality classes is a significant contribution with potential applicability to many different fields ranging from neuroscience, game theory, and economics.

Improvements and Detailed Suggestions:

  1. Aging Phenomenology in landscape models. On page 3, the authors claim that the aging phenomenology they observe "is significantly different from glassy dynamics on a rough landscape". This point needs further clarification. It is presumed that by "landscape" the authors refer to energy landscapes. In such a case, the interactions between species must be symmetric, and aging dynamics have already been shown in previous works, prior to Ref. [22]. The authors should elucidate how their approach differs from these earlier studies in which out-of-equilibrium aging dynamics occurring in the multiple equilibria regime were studied by using disordered system techniques.

  2. Parameter choice. On page 5, the choice of the parameter $\mu$ needs to be better clarified. I guess that the value 10 is chosen to avoid unbounded growth. The authors should comment more on the robustness and sensitivity of their results w.r.t. different values of the interaction strength.

  3. Correlation and noise fluctuations. At the end of the introduction, the authors present a plot of the correlation fluctuation, which does not decay to zero, but rather to a value near 0.2. Later, at the end of Sec. 3, they state that noise fluctuations completely decay to zero. I would appreciate a more detailed explanation of this aspect and clear evidence of the decay to a vanishing value. If this were not the case -- see for instance recent works for the mixed p-spin model where the correlation keeps memory of the initial condition — the perturbative approach they are presenting would break down and could require more complex techniques, such as Padé approximations. 

  4. Stationary Dynamics and Immigration Parameter. In section 4.1, the discussion on stationary dynamics as the immigration parameter tends to zero, is based on the (strong) assumption that the function g(s) cannot push solutions out of the confining region. This is inherently related to the model construction and the quadratic self-regulation term. In other realistic systems, such as tissue evolution or tumor growth, populations might not be confined but could blow up in certain regimes. The authors should comment on the applicability of their results to more realistic scenarios. Additionally, the two limits $s \rightarrow \infty$ and $z \rightarrow 0$ need careful handling: it is presumed that in the fluctuating phase they do not commute, hence necessitating a specific protocol.

  5. Technicality and Readability. Some sections, particularly 5.1 and 5.3, are very technical and difficult to follow for those not deeply familiar with the context and mathematical formalism. To facilitate the reader, I would recommend emphasising the physical significance and applicability of these results, moving extensive calculations to the appendix, and also providing a schematic overview or a summary table of the three regimes at the end.

  6. Finite-size communities. The issue of a finite number of species — crucial for understanding the emergence of crossovers and the phenomenology of artificial communities — is only briefly mentioned in one sentence of the conclusions. A more comprehensive commentary and generalisations of the findings in this direction would be appreciated.

  7. Numerics. The section on numerical methods is concise but could benefit from a more thorough explanation to ensure reproducibility. Expanding Appendix A to include a step-by-step explanation of the numerical methods used, along with discussions on assumptions, approximations, and potential limitations, would enhance this part.

To summarize, this paper is a valuable contribution to the field of population dynamics and phase transitions. Addressing the points raised above will enhance the paper’s clarity. The thorough theoretical approach, combined with more impactful explanations, would ensure that the paper reaches a wider audience in the scientific community.

Recommendation

Ask for minor revision

---

## Round 2 · Referee Report · Anonymous (Referee 2) · 2024-7-23

Report

I apologise sincerely that it has taken me so long to submit my report. This simply slipped off my radar.

I first address the journals "expectations" and then the "general acceptance criteria". Towards the end of my report, I add further comments to justify my reasoning.

EXPECTATIONS:

The authors have indicated that their paper meets the following expectations: - "Detail a groundbreaking theoretical/experimental/computational discovery" - "Present a breakthrough on a previously-identified and long-standing research stumbling block".

I agree that the work contains a significant theoretical and computational advance and discovery, and that it addresses an interesting question, namely the critical behaviour of random generalised Lotka-Volterra systems with immigration. In my view there is sufficient substance here to warrant publication.

I am unsure how to read SciPost's criteria though, words such as "groundbreaking" and "breakthrough" are very difficult to interpret. Does this include only things such as discovering the Higgs boson or room-temperature superconductivity, or are less spectacular results also covered. Going by existing publications in the journal, I assume that "groundbreaking" and "breakthrough" are mostly hyperbolic descriptions of what the SciPost is looking for. With this mind mind, I'd say that the criteria regarding substance are met.

GENERAL ACCEPTANCE CRITERIA:

  • "Be written in a clear and intelligible way, free of unnecessary jargon, ambiguities and misrepresentations"

This criterion is not met in my view. See below for further comments.

  • "Provide sufficient details (inside the bulk sections or in appendices) so that arguments and derivations can be reproduced by qualified experts"

This criterion is fulfilled.

  • "Provide citations to relevant literature in a way that is as representative and complete as possible"

This criterion is fulfilled. The authors could say in a little more detail (perhaps upfront) what is new here in relation to reference [21]. I do not doubt that there is significant novelty, but it should be made more clear (in an understandable manner) what exactly this is.

  • "Contain a clear conclusion summarizing the results (with objective statements on their reach and limitations) and offering perspectives for future work."

There is a summary of the main results at the beginning, but this is very difficult to understand. In this regard the criterion is only partially met. Avenues for future work are discussed in the conclusions, and I have no complaints about that.

  • "Contain a detailed abstract and introduction explaining the context of the problem and objectively summarizing the achievements"

Yes, but as stated above, the summary in Section 2 is rather difficult to understand.

  • "Provide (directly in appendices, or via links to external repositories) all reproducibility-enabling resources: explicit details of experimental protocols, datasets and processing methods, or processed data and code snippets used to produce figures, etc."

Yes, this is given. In some sense the main body of the paper contains too much detail in fact (see my comments below).

FURTHER COMMENTS:

There can be no doubt that this is significant and interesting work. My main concern is the presentation in the manuscript. I understand that the subject matter is very technical, and therefore it is difficult to present this to a an audience which goes beyond a very narrow community of experts. Nonetheless I think the authors need to do a little more work to make this more suitable to a broader set of readers.

Having a summary like that in Section 2 is a good idea. I am unsure though if this is best placed where it is, or perhaps better near the end (before the discussion). That is ultimately up to the authors. My main comment is that Section 2 itself is already very technical. The authors have made some attempt to systematise this, but still this remains somewhat convoluted and it is difficult to "see the forest for the trees". It would be good if the authors could do more to synthesise their results, and present things at a more coarse-grained level and to say what the significance of their findings is at a more non-technical level. In addition to modifying the text, perhaps some sort of diagram outlining the different regimes would help. Ideally, a general statistical physicist, who has not heard of the Lotka-Volterra system, should be able to get an idea what the main results are and why their are interesting/what they "mean". I know this is difficult to do, but nonetheless, I think it would be good if the authors could try.

Similarly, Sections 4 and in particular 5 are way too technical and contain too much detail. Don't get me wrong, these details should be made available to the expert reader, but for everyone else this is just confusing. The right place for this is an appendix or a supplement. In this way the main text (Sections 4 and 5) could be streamlined and focus on the actual physics, the main observations and messages. An interested but uninitiated reader should be able to understand what the main findings are without having to work through pages and pages of calculations. This part of the paper therefore needs an overhaul.

I am recommending a "major revision" because I think ideally Sections 2, 4 and 5 would look very different in an updated version of the manuscript. However, I stress that I am not asking the authors to do more actual work or to add substance to the manuscript. This is simply a matter of presenting these exciting findings in a form that can be understood and appreciated by more than a very narrow circle of experts.

Recommendation

Ask for major revision

---

## Editorial Decision

resubmitted